# A Theoretical Perspective for Speculative Decoding Algorithm

**Ming Yin**[*]
Princeton University
my0049@princeton.edu

**Minshuo Chen**
Northwestern University
minshuo.chen@northwestern.edu

**Kaixuan Huang**
Princeton University
kaixuanh@princeton.edu

**Mengdi Wang**
Princeton University
mengdiw@princeton.edu

## Abstract

Transformer-based autoregressive sampling has been the major bottleneck for slowing down large language model inferences. One effective way to accelerate inference is *Speculative Decoding*, which employs a small model to sample a sequence of draft tokens and a large model to validate. Given its empirical effectiveness, the theoretical understanding of Speculative Decoding is falling behind. This paper tackles this gap by conceptualizing the decoding problem via markov chain abstraction and studying the key properties, *output quality and inference acceleration*, from a theoretical perspective. Our analysis covers the theoretical limits of speculative decoding, batch algorithms, and output quality-inference acceleration tradeoffs. Our results reveal the fundamental connections between different components of LLMs via total variation distances and show how they jointly affect the efficiency of decoding algorithms.

## 1 Introduction

The recent surge of scaling Transformer models has led to the flourishing of AI, where success has been witnessed in wide areas such as natural language [39, 1], computer vision [12, 17], video generations [3, 19], and robotics [8, 31]. In the meantime, the decoding process also becomes more and more time-consuming as the model size scales up. This is mainly due to the autoregressive nature of Transformers, where each generated token also serves as the input for future generations. As a result, decoding $T$ tokens would take $T$ forward passes of the full model.

A recent effort to tackle this challenge is *speculative decoding* (SD) [10, 24], where the autoregressive sampling is performed on a small draft model and the large language model verifies tokens generated by draft model to decide whether it should be accepted/rejected. Once a token is rejected, the generation process will start from the most recently accepted token, until a full response is completed. Speculative decoding achieves 2-2.5$\times$ LLM inference speedup empirically, while preserving the quality of generation.

Subsequently, numerous studies [26, 25, 23, 46] have expanded this methodology, enabling further inference acceleration. Intuitively, for speculative decoding, when the generation distribution of small model $p$ and large model $q$ are close to each other, decoding is faster (since less rejection occurs), and when the distribution overlap between $p$ and $q$ is small, the opposite happens. However, a precise understanding of inference accelerating given the small model $p$ and large model $q$ remains elusive. This motivates us to ask the following question:

---

[*]Correspondence to: my0049@princeton.edu, mengdiw@princeton.edu.

38th Conference on Neural Information Processing Systems (NeurIPS 2024).

*What is the fundamental limit for inference acceleration via speculative decoding? In addition, what is the best trade-off between inference acceleration and output quality for speculative decoding?*

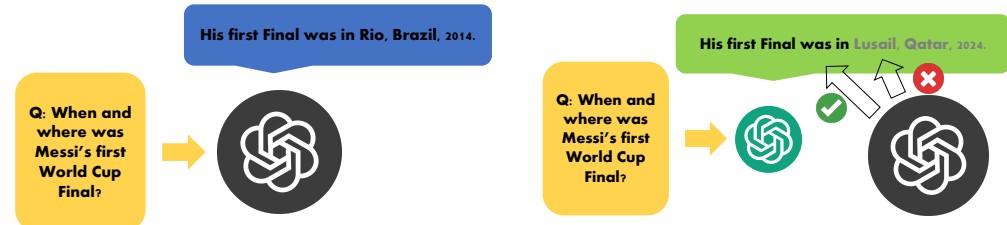

Figure 1: Left: Standard Auto-Regressive Decoding (Algorithm 3) v.s. Right: Speculative Decoding (Algorithm 1), where a large model is used to validate the responses of the small model.

In this paper, we answer these questions from the theoretical lens. Our contributions are summarized as follows.

We formalize the decoding problem through the Markov Chain abstraction that establishes the theoretical setup. We draw the connection **runtime = # rejections** and use it to measure efficiency. We derive the exact formula, fully characterized by distribution $p$ and $q$, for the expected rejections $\mathbb{E}[N_{\text{rej}}]$ for Speculative Decoding (Theorem 1). This renders a theoretical reference for understanding the acceleration rate $T/\mathbb{E}[N_{\text{rej}}]$.

Next, to understand whether Speculative Decoding can be further improved, we generalize it to a class of rejection-based algorithms 2 where probability $b_t$ and distribution $\mathcal{P}_t$ can be customized. We prove in Theorem 2 that any unbiased algorithm cannot have fewer rejections than Speculative Decoding. This indicates its optimality among the class, and having fewer rejections needs to suffer quality loss or requires extra information.

Furthermore, we consider a batch version of Speculative Decoding (Algorithm 4) that utilizes multiple draft sequences. We show our batch algorithm is unbiased. We derive the expected rejections that is fully expressed by $p$ and $q$ and exhibit the improvement over non-batch version in Theorem 3. We provide examples and detailed discussion to explain how our theory characterize the improvement.

In section 4, we shift from unbiased algorithms and study the tradeoff between *inference cost* and *quality degradation*. We formulate this into an optimization model (1). Theorem 5 established a linear Pareto front that characterizes the tradeoff between inference cost and quality degradation (Figure 4). A simple experiment in Section 4.2 is also consistent with our theoretical finding.

Last but not least, our technical results involve novel analysis, for instance, the design of $\mathcal{V}_+, \mathcal{V}_-$ in the lower bound proof C and the iterative computation for $f$ in D.1. They are the first of its kind and consist of our technical contributions. We provide a proof sketch section in Appendix A.

## 1.1 Related works

**Speculative Decoding and its applications.** Speculative execution, where performing speculative work can expedite computation on parallel machines by allowing certain tasks to commence before their necessity is confirmed, can date back to [9, 15]. Recently, [10, 24] formalize this idea with rejection sampling based design for LLM Decoding and achieve multiple-time inference acceleration compared to vanilla auto-regressive decoding. There are fruitful studies [41, 26, 25, 37, 23, 46, 32, 27, 18, 4, 34, 2, 30, 43, 11, 42, 45, 29, 35, 5, 44, 40, 20] since then, and they improve speculative decoding from different angles such as online updating [25], multiple candidates [45], retrieval technique [18], Multimodality [16] or even decoding without draft models [14, 6].

**Theoretical endeavor for Speculative Decoding.** There are also works that study the theoretical properties for speculative decoding (SD). In particular, [37] considers speculative decoding from the optimal transport perspective and show it is optimal in the single token regime. It further extends to the $k$ multiple draft token setting and formulates the optimal transport solution via linear programming

with exponential in $k$ computation time. An approximate sequential selection algorithm is also proposed with linear time. [2] further proposes the improved plan, and [36] extends [37] to the block-level optimal transport for SD. [34] investigates the synergy between draft length and batch size for Speculative Decoding and formulate the optimal speculation length as the root of a polynomial equation. [26] proposes the SpecInfer, a batch algorithm that uses small speculative models to form the token tree, and proves its output matches the distribution of the large model. [45] considers batch speculative decoding without replacement to avoid repeatedly sampling rejected tokens and proves it keeps the decoding quality. Nevertheless, the findings for inference acceleration of these works are mostly empirical, lacking theoretical guarantees.

---

**Algorithm 1** Speculative Decoding [10, 24]

1: **Input**: Set probability $b_t = \min\{1, \frac{q_t}{p_t}\}$ and the distribution $\mathcal{P}_t = [q_t - p_t]_+$ in Algorithm 2.
2: **Require**: $p_t(\cdot) := p_t(\cdot|x_{1:n-1}, \tilde{x}_{n:t-1})$, $q_t(\cdot) := q_t(\cdot|x_{1:n-1}, \tilde{x}_{n:t-1})$. $\forall t \geqslant n$.
3: **for** $t = n : T$ **do**
4:      Sample $r \sim \mathsf{Uniform}[0, 1]$.
5:      **if** $r \leqslant \min\left\{1, \frac{q_t(\tilde{x}_t)}{p_t(\tilde{x}_t)}\right\}$ **then**
6:          Accept with $x_n = \tilde{x}_t$. $n \leftarrow n + 1$.
7:      **else**
8:          Sample $x_n \sim [q_t - p_t]_+ (\cdot)$.
9:          $n \leftarrow n + 1$. Break.
10:          // Recall $[q_t - p_t]_+$ in Section 2.1
11:      **end if**
12: **end for**

**Algorithm 2** Framework for Rejection-based Decoding

1: **Init**: Horizon $T$, models $q_t, p_t$. Lookahead $K = T$. Prompt $x_0$. $n = 0$.
2: **Require**: Probability $b_t$, distribution $\mathcal{P}_t$.
3: **while** $n < T$ **do**
4:      **for** $t = n : T$ **do**
5:          Sample drafts $\tilde{x}_t \sim p_t(\cdot|x_{1:n-1}, \tilde{x}_{n:t-1})$.
6:      **end for**
7:      Obtain the target logits *in parallel* for $\tilde{x}_{n:T}$ as $q_n(\cdot|x_{1:n-1})$, $\ldots$, $q_T(\cdot|x_{1:n-1}, \tilde{x}_{n:T})$.
8:      **for** $t = n : T$ **do**
9:          Accept $x_n = \tilde{x}_t$ with prob. $b_t$. $n \leftarrow n + 1$.
10:          Else REJECTION:
11:          Sample $x_n \sim$ distribution $\mathcal{P}_t$. $n \leftarrow n + 1$. Break.
12:      **end for**
13: **end while**

---

## 2 Preliminaries

### 2.1 Background for decoding problems

In this section, we provide the mathematical formulation for decoding problems using Markov Chains, and we explain auto-regressive models and speculative decoding algorithm based upon that.

**A Markov Chain Model for Decoding.** We denote $x_0$ as the prompt.[2] $x_n$ is the $n$-th token output and $x_{1:T}$ is the trajectory output by the algorithm with $T$ to be the fixed decoding horizon.[3] Then any decoding algorithm can be characterized by a Markov Chain: state at $t$ is described by history $x_{0:t}$, the initial state is $x_0$; the state transition $P_t$ maps state $x_{0:t}$ to state $x_{0:t+1}$. In the context of decoding problem, the transition matrix is defined as $P(x_{0:t+1}|x_{0:t}) := p(x_{t+1}|x_{1:t})$. In particular, we use $p$ to denote the (conditional) distribution for the *draft model* that resembles small speculative model, and $q$ for the *target model* that represents large language model. We use $[q - p]_+$ to denote the normalized distribution for $\max\{0, q(x) - p(x)\}, \forall x \in \mathcal{V}$ with $\mathcal{V}$ being the token vocabulary. We also denote $\bar{\mathbb{E}}_{x \sim f}[g] := \sum_x f(x)g(x)$.

**Auto-Regressive Decoding.** Consider sampling a trajectory $x_{1:T}$ from an auto-regressive model, where for the given $x_{1:n-1}$, the next token $x_n$ follows the conditional distribution $q$. This decoding mechanism (Algorithm 3) is the prototype for Transformer-based LLMs (*e.g.* GPT-4). As mentioned in [29], the accuracy performance of Transformer-based LLMs has been shown to scale with model size, with larger models demonstrating improved capabilities [22]. However, this improvement comes at the cost of higher latency during inference and increased computational requirements.

**Speculative Decoding.** Different from large models, small models are usually much faster at the inference stage. Consequently, we can use a small model to perform auto-regressive sampling and assign large model as a verifier, where the goal of the large model is to check whether the

---

[2]We use the abstraction $x_0 := (y_1, y_2, \ldots, y_{n_{\text{prompt}}})$ with $y_i$'s being the tokens in the prompt.
[3]In general, $T$ is a random variable. However, we can append [EOS] tokens to keep the output length fixed.

token sampled by the small model should be accepted/rejected. Concretely, this procedure can be summarized into the following three steps:

- *Draft sampling*: given the verified tokens $x_{1:n-1}$, the draft model obtains $K$ sequential candidate tokens $\tilde{x}_{n:n+K-1}$ via sampling from $p(\cdot|x_{1:n-1}, \tilde{x}_{n:n+i})$, $i \in [K-1]$;

- *Conditional score computation*: given $\tilde{x}_{n:n+K-1}$, computing the logits of the $K$ tokens $q(\cdot|x_{1:n-1}, \tilde{x}_{n:n+i})$, $i \in [K-1]$ *in parallel*;

- *Token validation*: Accept the candidate token $\tilde{x}_t$ (as $x_n$) with some probability $0 \leqslant b_t \leqslant 1$. If accepted, continue to validate the next candidate $\tilde{x}_{t+1}$, otherwise reject and sample $x_n$ from some designed distribution $\mathcal{P}_t$.

The above process repeats until $x_{1:T}$ are generated, and the whole algorithm is summarized as the general rejection-based decoding 2. Specifically, *Speculative Decoding* [10, 24] designs $b_t(\tilde{x}_t) := \min\{1, \frac{q_t(\tilde{x}_t)}{p_t(\tilde{x}_t)}\}$ and distribution $\mathcal{P}_t(\cdot) := [q_t - p_t]_+(\cdot)$ (see Algorithm 1 and Figure 1).

## 2.2 Problem Setup

Speculative Decoding has two key merits. First, it maintains **output quality**, meaning the output distribution by the algorithm is identical to the output distribution of the large model $q$, and we term it *distribution unbiasedness*. Second, it has **fast inference** property since the auto-regressive sampling is performed on the small model and the target model only verifies. With (possibly) multiple draft tokens being accepted very round, Speculative Decoding can be much faster than direct decoding on large models, and its inference is bottlenecked by the parallel score computation $q(\cdot|x_{1:n-1}, \tilde{x}_{n:n+i})$, $i \in [T-n]$. To highlight this, we defined it as the oracle call and make an assumption based on that.

**Definition 1.** *We define one trigger of obtaining logits for $\tilde{x}_{n:T}$ in Step 7 of 2 as one **Oracle call**.*

**Assumption 1.** *We assume that: (1) compared to the large model, the computational cost of the draft/small model is negligible. (2) each oracle call has runtime $O(1)$.*

**Remark 1.** *We assume the above only for the theoretical cleanliness. With negligible draft model, the lookhead $K = T$ in Algorithm 2. In other words, given $x_{1:n-1}$, instead of sampling the next $K$ draft tokens $\tilde{x}_{n:n+K-1}$, we are allowed to sample until the end, i.e. $\tilde{x}_{n:T}$. In practice, Assumption 1(1) also holds true in many cases. One example of negligible-cost model $c \approx 0$ is n-gram models, and the empirical evidence in [24, 28] shows n-gram draft model speeds up inference pretty well. In addition, for summarization tasks where long sequences are likely to repeat, any draft model that reuses tokens from the context with a matching prefix, is also cost negligible. Assumption 1(2) is also a standard abstraction, since parallelization consumes (roughly) equal computation as for a single logit. It will consume more memory, but such aspect is beyond the scope of our theoretical study.*

**Metric for measuring efficiency.** Desired algorithms should preserve the output quality and be efficient for decoding. To formalize, our theory aims to study the following two quantities:

- *Inference acceleration*: the ratio between the inference time of decoding large model $q$ and the inference time of decoding the algorithm;

- *Output quality*: the distribution bias between the algorithm and the large model distribution $q$. An algorithm maintains the output quality if it is distribution unbiased.

**Rejections, and why consider it?** A third metric for decoding is the number of draft tokens being rejected. With more draft tokens being accepted, the fewer rejections would occur. Therefore, algorithms with large number of rejections are slower than those with fewer rejections. This means *Rejections* serves as an alternative metric for the inference speedups. Throughout the paper, we use *number of rejections* for measuring *inference acceleration*.

To further motivate why choosing Rejections is appropriate, we draw its connection to inference runtime. For Speculative Decoding, the runtime is dominated by the number of oracle calls (Definition 1). After each rejection (Step 10 of 2 or Step 7 of 1), there is one oracle call, thus we obtain

$$\text{inference time} = \text{\# oracle calls} = \text{\# rejections}.$$

Notice the runtime for auto-regressive decoding (3) is $T$, therefore we have the relation: **inference acceleration** $= T/\text{\# rejections}$. Based on this setup, we present our main results in next sections.

## 3 Analysis on Efficiency and Optimality for Speculative Decoding

We start with the following theorem at the beginning of the section. It covers output quality, which is measured by distribution bias, and expected number of rejections for speculative decoding. Its proof is deferred to Appendix B.

**Theorem 1.** *We have the following two results for Speculative Decoding.*

*(1) We define random variables $R_n \in \{0, 1\}$ that indicates whether the $n$-th token is rejected (with $1$ being rejected). Here rejection means Line 7 of Algorithm 1 is executed. Then, the total number of rejections $N_{rej} = \sum_{n=1}^{T} R_n$. For Speculative Decoding (here $\mathsf{TV}$ denote the TV distance):*

$$\mathbb{E}[N_{rej}] = \sum_{n=1}^{T} \mathbb{E}_{x_{1:n-1} \sim q}[\mathsf{TV}(p_n(\cdot|x_{1:n-1}), q_n(\cdot|x_{1:n-1}))].$$

*(2) The output distributions of Algorithm 1 and the target model $q$ are identical,* i.e. *for any output sequence $x_{1:T} \in \mathcal{V}^T$, the joint the distributions over $x_{1:T}$ satisfies: $\mathbb{P}^{SD}(x_{1:T}) = q(x_{1:T})$.*

The first part of Theorem 1, to our knowledge, is the first result that characterizes the expected rejection for speculative decoding a sequence of length $T$ using $p$ and $q$. The second part of Theorem 1 shows the distribution unbiasedness for SD, which has been presented in [10, 24]. There are three interesting indications:

- If $\mathbb{E}[\mathsf{TV}(p_n, q_n)(\cdot|x_{1:n-1})] = 0$ for all $n$, all tokens are accepted and the accelerate rate $= T$;

- If $\mathbb{E}[\mathsf{TV}(p_n, q_n)(\cdot|x_{1:n-1})] = 1$ for all $n$, all tokens are rejected and the accelerate rate $= 1$, *i.e.* all $T$ tokens are sampled from the large model $q$;

- In general, the accelerate rate for SD is $T/\sum_{n=1}^{T} \mathbb{E}[\mathsf{TV}(p_n, q_n)(\cdot|x_{1:n-1})]$.

**Remark 2.** *[24] derive the expected number of token generated per run of Speculative Decoding as $1/\mathbb{E}[\mathsf{TV}(p, q)]$ for $K = \infty$.[4] Their result equals $T/\sum_n^T \mathbb{E}[\mathsf{TV}(p_n, q_n)(\cdot|x_{1:n-1})]$ when $\mathbb{E}[\mathsf{TV}(p_n, q_n)(\cdot|x_{1:n-1})]$ is identical for all $n$, and this is due to their assumption that the acceptance rates $\beta$ are i.i.d.. In contrast, our guarantee holds for the case that allows the sequential dependence between different decoding steps.*

**Simulation.** We also provide a simulation of Speculative Decoding and compare it with our Theorem 1 in the left panel of Figure 2(a) with horizon $T = 50$, $p_n, q_n, n = 1, \ldots, 50$ are nonstationary Markov Chains. The green line is the empirical average rejections among $100 \times N$ runs of Algorithm 1 and the orange line the theoretical value computed via Theorem 1. From the simulation, after $5000$ runs the empirical average rejections converge to our theoretical value $16.41$. In this example, the acceleration rate is $50/16.41 = 3.05$. The specifications of the simulation is included in Appendix F.

### 3.1 Optimality of Speculative Decoding

Now we have analyzed the efficiency of speculated decoding and provided mathematical characterization of the number of expected rejections, which is depicted by the $\mathsf{TV}$ distance and scales linearly with the inference time. A natural next-step question to ask is: *Is there any other rejection-based algorithm 2 that can do better?* We answer this question in the next theorem.

**Theorem 2** (**Instance-dependent Rejection Lower Bound**). *For an instance $\mathcal{P} := (p, q)$, where $p, q$ stand for the distributions of the draft model and the target model respectively, defining the family of algorithms as $\mathcal{F} := \{\mathcal{A} : \mathcal{A}$ is a specification of Algorithm 2 that satisfies $\mathbb{P}_t^{\mathcal{A}} = q_t \; \forall t$ (i.e., unbiased)$\}$. For an algorithm $\mathcal{A}$, denote $N_{rej}$ as the number of rejections. Then we have the lower bound*

$$\inf_{\mathcal{A} \in \mathcal{F}} \mathbb{E}_{\mathcal{P}}^{\mathcal{A}}[N_{rej}] \geqslant \sum_{n=1}^{T} \mathbb{E}_{x_{1:n-1} \sim q}\left[\mathsf{TV}(p_n, q_n)(\cdot|x_{1:n-1})\right].$$

---

[4]This is from their equation (1) that has $1/(1 - \alpha)$ with $\alpha = \mathbb{E}(\min(p, q))$ and $1 - \mathbb{E}(\min(p, q)) = \mathbb{E}[\mathsf{TV}(p, q)]$.

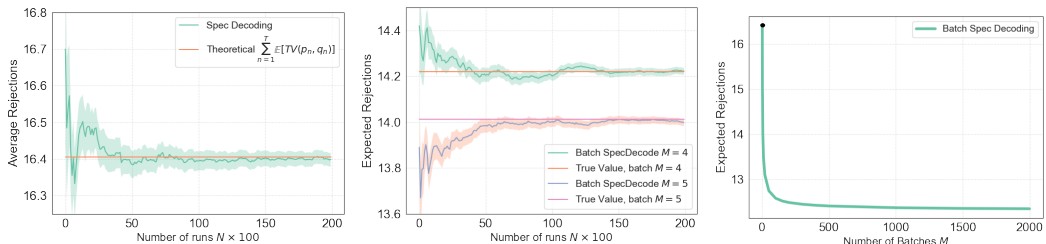

Figure 2: The numeric instance in this figure chooses $p, q$ to be nonstationary Markov Chains with horizon $T = 50$. Left ($a$): A simulation of Speculative Decoding. The green line is the empirical average rejections among $100N$ runs and the orange line the theoretical value computed via Theorem 1. Middle ($b$): Batch Speculative Decoding simulations with batch $M = 4, 5$. The green/purple lines are the empirical average rejections among $100N$ runs and the orange/pink lines are the theoretical values computed via Theorem 3. Right ($c$): The scaling law of expected rejections for Batch SD as a function of $M$. It converges to a limit as $M \to \infty$.

**Takeaways.** Via Theorem 2, the answer to the key question is: *No rejection improvement can be made in Algorithm 2 by changing the acceptance probability $b_t$ and the distribution $\mathcal{P}_t$ if we want to keep the distribution unbiasedness.* We point out that lower bound of Theorem 2 matches the complexity upper bound of Speculative Decoding given by Theorem 1. This result confirms that speculative decoding is optimal in the class of all rejection-based methods.

The practical implication is that there is no need to tweak acceptance probability, as it will not make performance better. In the next Section 3.2, we will see that Speculative Decoding can be provably improved, as long as one can decode and verify multiple sequence of tokens in parallel.

**Connection to optimal transport.** We mention [37] nicely consider maximizing the acceptance probability from the *optimal transport* perspective. For a single token, the optimal transport cost is $\sum_x \min(p(x), q(x))$, which corresponds to the optimal expected rejection $\mathsf{TV}(p, q)$. However, for the sequential $T$ tokens, their Section 5,6 does not provide a explicit formulation for optimal acceptance/rejections. In this sense, their optimality result can be cast as a special case of ours. However, we do emphasis there are differences in the settings, where [37] consider the optimal transport and we study the class $\mathcal{F}$.

## 3.2 Analysis for Batch Speculative Decoding

To further improve the provable efficiency, we consider batch algorithms that extend the speculative decoding with multiple candidate draft sequences. Most of the existing works [32, 34, 26, 45, 21] formulates batch sequences via a speculation tree structure. In particular, the representative work [26] propose the merged token tree structure that combines sequences with the same prefix. A *depth-first search* is designed for speculation to ensure the unbiasedness of the algorithm. In addition, [33] devise the parallel decoding structure that speculate the token of each responses given its previous tokens are accepted. Motivated by these works, we consider a batch version of speculative decoding algorithm using a simpler parallel structure (Left of Figure 3). Our Algorithm 4 can be viewed as a simplified approximation to those batch algorithms. There are several differences that distinguish batch algorithm from the non-batch version. We highlighting them as follows.

**Difference1: Oracle call.** At the beginning of batch speculation, $M$ draft sequences are generated in parallel as well as the corresponding logits $q$. This corresponds to Step 4-9 of Alg 4 and is defined as one *oracle call* which we assume to have unit computation $O(1)$;[5]

**Difference2: Speculation procedure.** It follows the *DFS* principle: If the first token of a response is accepted, only that response will be speculated until rejection. For instance in the Left panel of Figure 3, if the token 'deep' is accepted, the algorithm will keep verifying token 'learn', 'ing' until rejection and the rest of sequences won't be examined; if 'deep' is not verified, then the algorithm will keep examing 'rein'. In this case, rejection happens only if 'deep', 'rein' and 'atten' are all

---

[5]We mention batch drafting in parallel will cause more memory and arithmetic operations, but not computation time.

rejected. Once rejection happens, the process will restart. By this design, it still holds true that: *inference time = # oracle calls = # rejections.*

Again, we measure the output quality and rejections for the batch algorithm. The following main result (Theorem 3) provides the explicit characterization for rejections and batch improvement. Detailed proofs and discussions can be found in D.

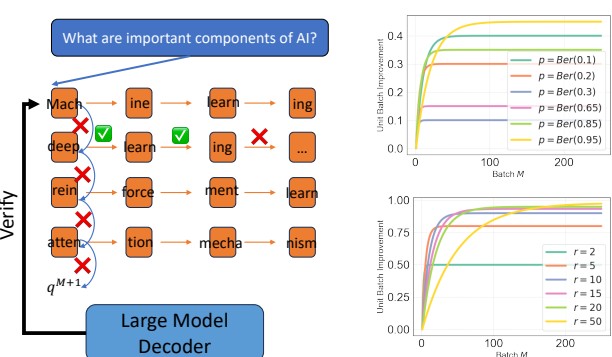

Figure 3: Left: Batch Speculative Decoding. Right: Batch Improvement vs. Batch size $M$. Upper: Bernoulli distributions with $q = Ber(0.5)$. Lower: $p \sim \mathrm{Unif}(V)$, $q \sim \mathrm{Unif}(V')$ with $r = V/V'$.

**Theorem 3** (**Unbiasedness and efficiency of batch SD**). *Recall the definition of $R_n$ and $N_{rej}$ in Thm 1, and iteratively define: $q^{m+1} = [q^m - p]_+$, $\forall m \in [M]$ with $q^1 = q$ being the target distribution. Then, for Batch Speculative Decoding 4, $\mathbb{P}^{Batch}(x_{1:T}) = q(x_{1:T})$. $\forall x_{1:T} \in \mathcal{V}^T$. Moreover,*

$$\mathbb{E}[N_{rej}] = \sum_{n=1}^{T} \mathbb{E}_{x_{1:n-1} \sim q}[\mathsf{TV}[q,p](\cdot|x_{1:n-1})] - \underbrace{\sum_{n=1}^{T} \bar{\mathbb{E}}_{x_{1:n-1} \sim f}[\mathsf{TV}(q,p)(x_{1:n-1}) - [\prod_{m=1}^{M} \mathsf{TV}(q^m,p)(x_{1:n-1})]]}_{\text{Batch Improvement}}$$

*where $f(x_{1:n}) := \mathbb{P}(x_{1:n} \cap \{n\text{-th draft token rejected}\})$. $f$ can be iteratively computed via $p, q$.*

**Takeaways.** The expected rejection of Batch Decoding composes of two parts: *(i)* $\sum_{n}^{T} \mathbb{E}_q[\mathsf{TV}[q,p]]$ which is the rejection that matches the non-batch speculative decoding; *(ii)* the batch improvement (BI) which comes from our design and is always non-negative. To better understand how the batch improvement scale with $M$, we instantiate the single token $\mathrm{BI}(q,p) := \mathsf{TV}[q,p] - \prod_{m=1}^{M} \mathsf{TV}[q^m,p]$ with two simple examples.

**Uniform Distribution.** For the whole space $\mathcal{V}$, let $p$ be a uniform distribution with support $\mathcal{V}$ and $q$ be a uniform distribution over a subset of $\mathcal{V}$ with size $V' < V$. Let $V/V' = r$, in this case $\mathrm{BI}(\mathrm{Unif}(V'), \mathrm{Unif}(V)) = (1 - \frac{1}{r}) - (1 - \frac{1}{r})^M$, $r = V/V'$. Its pattern is visualized in the lower right panel of Figure 3. The improvement converges to $1 - 1/r$ and is always positive as long as batch size is at least 2. Also, when the vocabulary size $V$ scales up, *i.e.* $r$ goes up, the improvement is going to zero. This is not surprising, since the probability of draft sample to fall within in the support of target distribution is very small.

**Bernoulli Distribution.** Suppose $u \geq v$ and let $p \sim \mathrm{Ber}(u)$, $q \sim \mathrm{Ber}(v)$. Then $\mathrm{BI}(\mathrm{Ber}(v), \mathrm{Ber}(u)) = |u - v| \cdot (1 - u^{M-1})$. Its pattern is exhibited in the upper right panel of Figure 3. Notice for both cases, the limiting batch improvement is $\mathsf{TV}(p,q)$, which is only significant when $p$ deviates from $q$. This provides the practical design heuristic: *batch design can significantly reduce the number of rejections when the draft distribution $p$ and target distribution $q$ are different from each other and won't make too much improvement when $p$ and $q$ are close.*

**Batch Simulation.** The Middle panel of Figure 2(b) shows Batch Speculative Decoding simulations with batch $M = 4, 5$. The green/purple lines are the empirical average rejections simulated via Algorithm 4 and the orange/pink lines are the theoretical values computed via Theorem 3. The right panel of Figure 2(c) plots the rejection vs. batch size. In particular, the the black dot with coordinate $(1, 16.41)$ represents the Speculative Decoding 1. The numeric example shows when $M \to \infty$, $\mathbb{E}[N_{rej}] \nrightarrow 0$. Intuitively, this is partial due to, if the distribution mismatch between $p$ and $q$ is large,

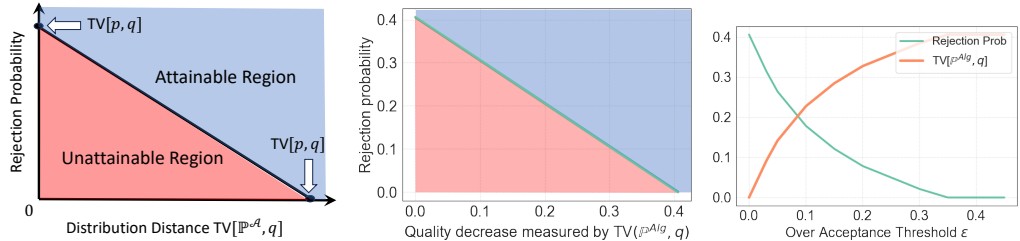

Figure 4: Left $(a)$: The Pareto Front between *Rejection Probability* $\mathbb{P}^{\mathcal{A}}$(reject) vs. *Distribution bias* TV$[\mathbb{P}^{\mathcal{A}}, q]$. For a given rejection probability, the black line denotes the optimal deviation Loss$^*_{\text{TV}}$. Middle $(b)$ and Right $(c)$: A numeric example. In the plot, the over acceptance $\epsilon$'s are set as positive constants that define $b(x) = \min\{1, \frac{q(x)+\epsilon}{p(x)}\}$.

rejection is unavoidable no matter how many draft responses are sampled from $p$. We also formally prove this in Proposition 1. This theoretical discovery indicates that having a very large draft batches does not necessarily result in significant inference speedups compared to small batches.

# 4 Analysis on the Optimal Rejection-Distribution Bias Tradeoff

In earlier sections, we focus on analyzing SD or Batch SD which are distribution unbiased. To further reduce rejections, the algorithm may have to compromise on output quality. Nevertheless, it is usually acceptable for many real-world applications. For instance, for the family of Gemini models [38], Google may deploy the small model *Gemini Nano* as the draft model $p$ and *Gemini Ultra* as the target model $q$ and tune the acceptance probability $b$ (in Algorithm 2) higher such that *Gemini Nano* is applied more often for the purpose of less expenditure. Therefore, an intriguing question to ask is: *For biased algorithms, what is the optimal tradeoff between rejection and distribution bias?*

## 4.1 An Optimization Formulation and Pareto-optimal Characterization

We measure the distribution bias between $\mathbb{P}^{\mathcal{A}}$ and $q$ via TV distance.[6] Concretely, for any algorithm $\mathcal{A} := (b, \mathcal{P})$ in 2 with acceptance probability $b$ and rejection distribution $\mathcal{P}$, we fix $b$ and aim to optimize the following objective

$$\text{Loss}^*_{\text{TV}}(b) := \min_{\mathcal{P}} \text{TV}[\mathbb{P}^{\mathcal{A}}, q], \quad where \; \mathcal{A} := (b, \mathcal{P}). \tag{1}$$

We wish to solve the above since it characterizes the minimal distribution bias for any design $b$. We present our solution in the next.

**Theorem 4** (Optimal solution to optimization (1)). *We can show the objective (1) is equivalent to*

$$Loss^*_{\text{TV}}(b) := \frac{1}{2} \min_{\mathcal{P}} \sum_x \left| q(x) - b(x)p(x) - \mathcal{P}(x) \sum_{\tilde{x}} [1 - b(\tilde{x})]p(\tilde{x}) \right| \; s.t. \; \sum_x \mathcal{P}(x) = 1, \; \mathcal{P}(x) \geqslant 0, \; \forall x.$$

*Suppose $\sum_x [1 - b(x)]p(x) > 0$. Define $A(x) := \frac{q(x) - b(x)p(x)}{\sum_{\tilde{x}} [1 - b(\tilde{x})]p(\tilde{x})}$, and the sets $A_+ = \{x \in \mathcal{V} : A(x) \geqslant 0\}$, $A_- = \{x \in \mathcal{V} : A(x) < 0\}$. Then the set of optimal distributions of objective (1) is characterized as $\{\mathcal{P}^* : \mathcal{P}^*|_{A_-}(\cdot) = 0; 0 \leqslant \mathcal{P}^*|_{A_+}(\cdot) \leqslant A(\cdot)\}$, and the optimal value is $Loss^*_{\text{TV}}(b) = \frac{1}{2} \sum_x |q(x) - b(x)p(x)| - \frac{1}{2} \sum_x (1 - b(x))p(x) \geqslant 0$.*

Theorem 4 is a universal characterization that contains both biased and unbiased situations. 1. If $b$ is less than Speculative Decoding threshold, *i.e.* $b \leqslant \min\{1, q/p\}$ then $A$ becomes a probability distribution and the optimal distribution $\mathcal{P}^*$ equals $A$ which is also unbiased (Loss$^*_{\text{TV}}(b) = 0$); 2. If $b$ exceeds the Speculative Decoding threshold, then $A$ is no longer a distribution and there are multiple optimal distributions $\mathcal{P}^*$. In this case, the optimal distribution bias Loss$^*_{\text{TV}}(b) > 0$ for Algorithm 2.

---

[6]We mention TV distance is only one metric for measuring distribution bias. In general, any distribution distance can be considered. We leave a thorough study for all other distances as future work.

**Main Takeaways.** Via Theorem 4, we derive the pareto front (the optimal tradeoff) between rejection probability[7] $\mathbb{P}(\text{reject})$ vs. distribution distance $\mathsf{TV}[\mathbb{P}^{\mathcal{A}}, q]$ (Left panel of Figure 4). The blue region can be realized by some algorithm 2, and the red region cannot. $(0,0)$ is the "perfect algorithm" (no rejection, no bias) which does not exists, and, in particular, $(0, \mathsf{TV}[p,q])$ stands for Speculative Decoding. Surprisingly, the pareto front is a straight line connecting $(0, \mathsf{TV}[p,q])$ and $(\mathsf{TV}[p,q], 0)$, which represents a linear relationship between the rejection probability and the optimal $\mathsf{TV}$ deviation. This is guaranteed by the following theorem.

**Theorem 5** (**Pareto front**). *For any Algorithm $\mathcal{A}$ in the class 2 that satisfies $\min\{1, \frac{q(x)}{p(x)}\} \leqslant b(x) \leqslant 1, \ \forall x \in \mathcal{V}$. Then*

$$\mathbb{P}^{\mathcal{A}}(reject) + Loss^*_{\mathsf{TV}}(b) = \mathsf{TV}[p,q].$$

*Here $\mathbb{P}^{\mathcal{A}}(reject) = 1 - \sum_x b(x)p(x)$ and $Loss^*_{\mathsf{TV}}(b) := \min_{\mathcal{P}} \mathsf{TV}[\mathbb{P}^{\mathcal{A}}, q]$.*

We plot the numerical example using two Markov Chains in the middle and right panel of Figure 4 that coincides with our theoretical finding. In the figure, the acceptance probability is set to be $b(x) = \min\{1, \frac{q(x)+\epsilon}{p(x)}\}$. The orange line in $(c)$ and the green boundary in $(b)$ are computed via $Loss^*_{\mathsf{TV}}(b)$ from Theorem 4. The complete proofs for Theorem 5 and Theorem 4 are deferred to Appendix E. For the clearness of illustration, we focus on the pareto front between rejection vs. the minimal $\mathsf{TV}$ deviation for a single token.

## 4.2 Experiment

We provide an additional simple experiment to show the effectiveness of our Pareto-optimal solution in Theorem 4. Consider objective (1). For the given acceptance probability $b > \min\{1, q/p\}$, we have two options for $\mathcal{P}$: 1. Baseline: select $\mathcal{P}$ to be suboptimal to (1), *i.e.* simply set $\mathcal{P} := q$, which is the target distribution; 2. Set $\mathcal{P} := \mathcal{P}^*$ be the optimal distribution of Theorem 4. We call the first method `Decoding-UNO` and the latter one `Decoding-OPT`. Instead of TV distance, we measure the quality via WinRate in our experiment. Concretely, for each prompt, we let `Decoding-UNO` and `Decoding-OPT` to generate responses independently, and use score models to compare whose generation has higher quality. We specify draft model $p$ as `pythia-70m` and target model $q$ as `pythia-2.8b` from EleutherAI [7]. We apply the score model to be `RM-Mistral-7B` or `GPT-4`. We test 200 prompts from `Alpaca-Farm-Eval` Dataset [13] with 500 responses/comparisons per prompt. Table 1 shows that `Decoding-OPT` achieves better performance than `Decoding-UNO` across different choice of $\epsilon$'s. Due to space constraint, the missing details are deffered to Appendix G.

| Method | RM-Mistral-7B | | | | GPT-4 | | | |
|---|---|---|---|---|---|---|---|---|
| | $\epsilon = 0.1$ | $\epsilon = 0.2$ | $\epsilon = 0.4$ | $\epsilon = 0.8$ | $\epsilon = 0.1$ | $\epsilon = 0.2$ | $\epsilon = 0.4$ | $\epsilon = 0.8$ |
| Decoding-OPT | 53% | 53.5% | 57.5% | 52.5% | 54.5% | 53% | 53.5% | 55% |
| Decoding-UNO | 47% | 46.5% | 42.5% | 47.5% | 45.5% | 47% | 46.5% | 45% |

Table 1: WinRate for `Decoding-OPT` vs `Decoding-UNO` with different over-acceptance threshold $\epsilon$. The acceptance probability $b(x) = \min\{1, \frac{q(x)+\epsilon}{p(x)}\}$.

## 5 Discussions

**On the optimality for batch algorithms.** Unlike their non-batch counterparts, our batch algorithm studies do not come with optimality guarantees. This is largely due to the diverse and arbitrary nature of batch algorithm designs, making it challenging to define a comprehensive class that encompasses a wide range of batch algorithms. While [37] investigate optimal batch algorithms through an optimal transport lens, their work does not extend to calculating optimal rejection rates or developing an efficient algorithm to achieve this (they only propose an approximate solution). Consequently, the pursuit of batch optimality remains an open field. Identifying the optimal batch algorithm could yield valuable insights for enhancing practical applications in real-world scenarios.

---

[7]Here the rejection probability is computed as $\mathbb{P}(\text{reject}) = \sum_{\tilde{x}} \mathbb{P}(\text{reject}, \tilde{x}) = \sum_{\tilde{x}} \mathbb{P}(\text{reject}|\tilde{x})\mathbb{P}(\tilde{x}) = \sum_{\tilde{x}}(1 - b(\tilde{x}))p(\tilde{x})$.

**Extending Speculative Decoding to other studies.** Speculative Decoding is a generic sampling approach that extends beyond mere decoding tasks. It holds potential for wider applications such as search engines and recommendation systems, where it can be employed to quickly generate and refine search outcomes or content suggestions, enhancing the overall efficiency and user experience of these systems. We leave these as future works.

## Acknowledgments

The authors would like to thank anonymous reviewers for their valuable feedback. Mengdi Wang acknowledges the support by NSF IIS-2107304, NSF CPS-2312093, and ONR 1006977.

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

# Appendix

## A Proof Sketch

### A.1 Proof sketch of Theorem 2.

For any algorithm $\mathcal{A} \in \mathcal{F}$, its design $b_n$ can be written as a function of any sequence $x_{1:n-1}, \tilde{x}_n$ with $0 \leqslant b_n(\tilde{x}_n, x_{1:n-1}) \leqslant 1.$[8] Based on this, we can further define the new function $\epsilon_n : \mathcal{V} \times \mathcal{V}^{n-1} \mapsto \mathbb{R}$ according to the following equation:

$$b_n(\tilde{x}_n, x_{1:n-1}) = \min\left\{1, \frac{q_n(\tilde{x}_n|x_{1:n-1}) + \epsilon_n(\tilde{x}_n, x_{1:n-1})}{p_n(\tilde{x}_n|x_{1:n-1})}\right\}. \tag{2}$$

Indeed, we can choose $\epsilon_n := b_n \cdot p_n - q_n$, and the validity of the definition is guaranteed by the Lemma 1. Then, the acceptance probability $b_n > \min\{1, \frac{q_n}{p_n}\}$ implies $\epsilon_n > 0$.

Next, we show for any $\mathcal{A} \in \mathcal{F}$, it must satisfy $b_n \leqslant \min\{1, \frac{q_n}{p_n}\}$ for all $n$. To this end, we design the two token sets $\mathcal{V}_+ = \{x : \exists n \text{ s.t. } \epsilon_n(x) > 0\}$ and $\mathcal{V}_- = \{x : \exists n \text{ s.t. } \epsilon_n(x) \leqslant 0\}$ and prove $\mathcal{V}_+ = \varnothing$, $\mathcal{V}_- = \mathcal{V}$.

Finally, by Lemma 2 in Appendix, any algorithm satisfies $b_n \leqslant \min\{1, \frac{q_n}{p_n}\}, \forall n \in [T]$ must have $\mathbb{E}_{\mathcal{P}}^{\mathcal{A}}[N_{\mathrm{rej}}] \geqslant \sum_{n=1}^{T} \mathbb{E}_q[\mathsf{TV}(p_n, q_n)(\cdot|x_{1:n-1})]$. Since $\mathcal{A} \in \mathcal{F}$ is arbitrary, this concludes the proof. The full proof is included in C.

### A.2 High Level Proof Sketch for the second part of Theorem 3

The derivation for the number of expected rejections using the batch speculative decoding is more involved than the Algorithm 4 due to the parallel response structure. The key step is to compute the intermediate quantity $\mathbb{P}^{\mathcal{A}}(\tilde{x}_n \text{ acc}, \tilde{x}_n = x_n|x_{1:n-1})$. Let $\tilde{x}_{n-1} \sim p(\cdot|x_{1:n-2})$, then there are two cases: $\tilde{x}_{n-1}$ accepted or rejected. We have

$$\mathbb{P}^{\mathcal{A}}(\tilde{x}_n \text{ acc}, \tilde{x}_n = x_n|x_{1:n-1}) = \mathbb{P}^{\mathcal{A}}(\tilde{x}_n \text{ acc}, \tilde{x}_n = x_n, \tilde{x}_{n-1}\text{acc}|x_{1:n-1}) + \mathbb{P}^{\mathcal{A}}(\tilde{x}_n \text{ acc}, \tilde{x}_n = x_n, \tilde{x}_{n-1}\text{rej}|x_{1:n-1})$$

$$= \underbrace{\mathbb{P}^{\mathcal{A}}(\tilde{x}_n \text{ acc}, \tilde{x}_n = x_n|\tilde{x}_{n-1}\text{acc}, x_{1:n-1})}_{p_a} \mathbb{P}^{\mathcal{A}}(\tilde{x}_{n-1}\text{acc}|x_{1:n-1})$$

$$+ \underbrace{\mathbb{P}^{\mathcal{A}}(\tilde{x}_n \text{ acc}, \tilde{x}_n = x_n|\tilde{x}_{n-1}\text{rej}, x_{1:n-1})}_{p_b} \mathbb{P}^{\mathcal{A}}(\tilde{x}_{n-1}\text{rej}|x_{1:n-1})$$

$$\tag{3}$$

In the process of finding $p_b$, we need to compute the the quantity $f(x_{1:n}) := \mathbb{P}(x_{1:n} \cap \{n\text{-th draft token rejected}\})$ and it can be recursively computed via (22) using $p, q$.

### A.3 Proof sketch for Theorem 4

Due to space constraint, we only summarize the high-level proof ideas for Theorem 4. Since $\sum_x A(x) = 1$, the original objective (1) can be equivalently rewritten as

$$\min_{\mathcal{P}} \frac{1}{2}\sum_x |A(x) - \mathcal{P}(x)|, \quad s.t. \quad \sum_x \mathcal{P}(x) = 1, \quad \mathcal{P}(x) \geqslant 0, \ \forall x \in \mathcal{V}. \tag{4}$$

We now find the solution of objective (4) in two steps.

**Step1:** Recall $A_+ = \{x \in \mathcal{V} : A(x) \geqslant 0\}$ and $A_- = \{x \in \mathcal{V} : A(x) < 0\}$, then any optimal $\mathcal{P}^*$ must satisfy $\mathcal{P}^*(x) = 0$ for all $x \in A_-$. This can be shown by contradiction via an alternative construction $\mathcal{P}'$ to reason $\mathcal{P}^*$ is suboptimal to $\mathcal{P}'$.

**Step2:** We characterize the optimal solutions of the objective. By Step1, we can show any optimal solution $\mathcal{P}^*$ satisfies $\sum_x |A(x) - \mathcal{P}^*(x)| \geqslant \|A\|_1 - 1$ and the equal sign can be achieved. Then we can convert $\|A\|_1 - 1$ back to $\mathrm{Loss}_{\mathsf{TV}}^*$. This also helps identify the optimal set of $\mathcal{P}^*$.

---

[8]It needs to follow $b_n \in [0, 1]$ since $b_n$ is a probability.

---

**Algorithm 3** Auto-Regressive Model Decoding

---

1: **Init**: Horizon $T$. Distribution $q$. Prompt $x_0$. $n = 0$.
2: **while** $n < T$ **do**
3:     Sample $x_n \sim q(\cdot|x_{1:n-1})$. $n \leftarrow n + 1$.
4: **end while**

---

## B  Proof of Theorem 1

**Theorem 6** (Restatement of the first part of Theorem 1). *We define random variables $R_n \in \{0, 1\}$ indicating whether the $n$-th token is rejected with $1$ being rejected (here rejection means Line 6 of Algorithm 1 is executed). Then, the total number of rejections $N_{rej} = \sum_{n=1}^{T} R_n$. For Speculative Decoding,*

$$\mathbb{E}[N_{rej}] = \sum_{n=1}^{T} \mathbb{E}_{x_{1:n-1} \sim q}[\mathsf{TV}(p_n(\cdot|x_{1:n-1}), q_n(\cdot|x_{1:n-1}))].$$

*Here $\mathsf{TV}$ denote the TV distance between two distributions.*

*Proof.* Given the verified the tokens $x_{1:n-1}$, we first compute $\mathbb{P}(\text{Reject at } n|x_{1:n-1})$. Denote the candidate draft token $\tilde{x} \sim p_n(\cdot|x_{1:n-1})$, then by law of total probability

$$\mathbb{P}(\text{Reject at } n|x_{1:n-1}) = \sum_{\tilde{x}} \mathbb{P}(\text{Reject at } n, \tilde{x}|x_{1:n-1})$$

$$= \sum_{\tilde{x}} \mathbb{P}(\text{Reject } \tilde{x}|\tilde{x}, x_{1:n-1})\mathbb{P}(\tilde{x}|x_{1:n-1})$$

$$= \sum_{\tilde{x}} \mathbb{P}(\text{Reject } \tilde{x}|\tilde{x}, x_{1:n-1})p_n(\tilde{x}|x_{1:n-1})$$

$$= \sum_{\tilde{x}} (1 - \min\{1, \frac{q_n(\tilde{x}|x_{1:n-1})}{p_n(\tilde{x}|x_{1:n-1})}\})p_n(\tilde{x}|x_{1:n-1})$$

$$= \sum_{\tilde{x}} \max\{0, p_n - q_n\}(\tilde{x}|x_{1:n-1}) = \mathsf{TV}(p_n, q_n)(\cdot|x_{1:n-1}),$$

where the third equal sign uses draft token is sampled from $p_n$ and the fourth equality is by design of Speculative Decoding (Algorithm 1, Line 4).

Lastly, by law of total expectation and above

$$\mathbb{E}[N_{rej}] = \sum_{t=1}^{T} \mathbb{E}[R_t] = \sum_{t=1}^{T} \mathbb{E}\left[\mathbb{E}[R_t|x_{1:t-1}]\right]$$

$$= \sum_{t=1}^{T} \sum_{x_{1:n-1}} \mathbb{E}[R_t|x_{1:t-1}]\mathbb{P}(x_{1:n-1})$$

$$= \sum_{t=1}^{T} \sum_{x_{1:n-1}} \mathbb{E}[R_t|x_{1:t-1}]q(x_{1:n-1})$$

$$= \sum_{t=1}^{T} \sum_{x_{1:n-1}} \mathbb{P}(\text{Reject at } n|x_{1:n-1})q(x_{1:n-1})$$

$$= \sum_{n=1}^{T} \mathbb{E}_{x_{1:n-1} \sim q}[\mathsf{TV}(p_n(\cdot|x_{1:n-1}), q_n(\cdot|x_{1:n-1}))].$$

Here the fourth equal sign comes from Speculative Decoding keep the distribution $q$ (Proposition **??**), the fifth equal sign comes from the event $\{R_n = 1\} = \{\text{Reject at } n\}$.  □

**Theorem 7** (Restatement of the second part of Theorem 1). *The output distributions of Speculative Decoding Algorithm 1 and the large model $q$ are identical, i.e. for any output sequence $x_{1:T} \in \mathcal{V}^T$, the joint the distributions over $x_{1:T}$ satisfies: $\mathbb{P}^{SpecDecoding}(x_{1:T}) = q(x_{1:T})$.*

**Remark 3.** *The proof of Theorem 7 is very similar to [10] except we have $K = \infty$. In addition, [10] proves the distribution match for a single token, we complement the proof to show the distribution match holds for a sequence of tokens $x_{1:T}$.*

*Proof.* In particular, we use induction to show the stronger result that $\forall\, t \in [T]$, $\forall x_1, x_2, \ldots, x_t \in \mathcal{V}$, it holds $\mathbb{P}_t^{\mathcal{A}}(x_1, x_2, \ldots, x_t) = q_t(x_1, x_2, \ldots, x_t)$.

**Step1:** Since $x_0$ is the prompt, its distribution is independent to $p, q$. Then for $t = 1$, applying Theorem 1 of [10] (with $K = \infty$) directly with $p$ and $q$ to be conditional distributions $p_1(\cdot|x_0)$ and $q_1(\cdot|x_0)$ gives $\mathbb{P}_1^{\mathcal{A}}(x_1|x_0) = q_1(x_1|x_0)$, which further implies $\mathbb{P}_1^{\mathcal{A}}(x_1) = q_1(x_1)$ (since the distribution of $x_0$ is independent of $p, q$).

**Step2:** assume $\mathbb{P}_t^{\mathcal{A}}(x_1, x_2, \ldots, x_t) = q_t(x_1, x_2, \ldots, x_t)$, we first show $\mathbb{P}^{\mathcal{A}}(x_{t+1}|x_{1:t}) = q(x_{t+1}|x_{1:t})$. Indeed, let $\tilde{x}_{t+1} \sim p(\cdot|x_{1:t})$, then by law of total probability

$$\begin{aligned}
\mathbb{P}^{\mathcal{A}}(x_{t+1}|x_{1:t}) =\, &\mathbb{P}^{\mathcal{A}}(\tilde{x}_{t+1} = x_{t+1}|x_{1:t})\mathbb{P}(\tilde{x}_{t+1}\text{ acc}|\tilde{x}_{t+1} = x_{t+1}, x_{1:t}) \\
&+ \mathbb{P}^{\mathcal{A}}(\tilde{x}_{t+1}\text{ rej}|x_{1:t})\mathbb{P}^{\mathcal{A}}(x_{t+1}|\tilde{x}_{t+1}\text{ rej}, x_{1:t})
\end{aligned} \tag{5}$$

By Algorithm 1,

$$\begin{aligned}
&\mathbb{P}^{\mathcal{A}}(\tilde{x}_{t+1} = x_{t+1}|x_{1:t})\mathbb{P}^{\mathcal{A}}(\tilde{x}_{t+1}\text{ acc}|\tilde{x}_{t+1} = x_{t+1}, x_{1:t}) \\
=\, &p(x_{t+1}|x_{1:t})\min\left(1, \frac{q(x_{t+1}|x_{1:t})}{p(x_{t+1}|x_{1:t})}\right) = \min\{p(x_{t+1}|x_{1:t}), q(x_{t+1}|x_{1:t})\}.
\end{aligned} \tag{6}$$

Next, the probability of rejection is:

$$\begin{aligned}
\mathbb{P}(\tilde{x}_{t+1}\text{ rej}|x_{1:t}) &= 1 - \mathbb{P}(\tilde{x}_{t+1}\text{ acc}|x_{1:t}) = 1 - \sum_{x'}\mathbb{P}(\tilde{x}_{t+1} = x', \ \tilde{x}_{t+1}\text{ acc}|x_{1:t}) \\
=\, &1 - \sum_{x'}\min\{p(x'|x_{1:t}), q(x'|x_{1:t})\} = \sum_{x'}\max\{0, q(x'|x_{1:t}) - p(x'|x_{1:t})\}
\end{aligned} \tag{7}$$

where the second equal sign comes from (6). Lastly, by the construction of the algorithm,

$$\mathbb{P}^{\mathcal{A}}(x_{t+1}|\tilde{x}_{t+1}\text{ rej}, x_{1:t}) = \frac{\max\{0, q(x_{t+1}|x_{1:t}) - p(x_{t+1}|x_{1:t})\}}{\sum_{x'}\max\{0, q(x'|x_{1:t}) - p(x'|x_{1:t})\}}. \tag{8}$$

Combining (8) and (7) yields

$$\mathbb{P}^{\mathcal{A}}(\tilde{x}_{t+1}\text{ rej}|x_{1:t})\mathbb{P}^{\mathcal{A}}(x_{t+1}|\tilde{x}_{t+1}\text{ rej}, x_{1:t}) = \max\{0, q(x_{t+1}|x_{1:t}) - p(x_{t+1}|x_{1:t})\}.$$

Plugging (6) and the above equation into (5) to obtain

$$\mathbb{P}^{\mathcal{A}}(x_{t+1}|x_{1:t}) = \min\{p(x_{t+1}|x_{1:t}), q(x_{t+1}|x_{1:t})\} + \max\{0, q(x_{t+1}|x_{1:t}) - p(x_{t+1}|x_{1:t})\} = q(x_{t+1}|x_{1:t}).$$

Finally, applying the above we obtain

$$\mathbb{P}_{t+1}^{\mathcal{A}}(x_{1:t+1}) = \mathbb{P}_t^{\mathcal{A}}(x_{1:t}) \cdot \mathbb{P}^{\mathcal{A}}(x_{t+1}|x_{1:t}) = \mathbb{P}_t^{\mathcal{A}}(x_{1:t}) \cdot q(x_{t+1}|x_{1:t}) = q_{t+1}(x_{1:t+1}),$$

where the last equal sign uses the induction hypothesis. $\square$

## C Lower Bound

**Theorem 8** (Restatement of Theorem 2). *Define the arbitrary instance $\mathcal{P} := (p, q)$, and define the family of algorithms as*

$$\mathcal{F} := \{\mathcal{A} : \mathcal{A} \text{ is a specification of Algorithm 2 that satisfies } \mathbb{P}_t^{\mathcal{A}} = q_t \; \forall t \; (\text{i.e., distribution unbiased})\}.$$

*For an algorithm $\mathcal{A}$, denote $N_{rej}$ as the number of rejections. Then we have*

$$\inf_{\mathcal{A} \in \mathcal{F}} \mathbb{E}_{\mathcal{P}}^{\mathcal{A}} [N_{rej}] \geqslant \sum_{n=1}^{T} \mathbb{E}_{x_{1:n-1} \sim q} [\mathsf{TV}(p_n, q_n)(\cdot | x_{1:n-1})] := \mathfrak{C}(\mathcal{P}).$$

**Remark 4.** *Theorem 8 shows the rejections of Algorithm 1 is tight at the instance level (over the family of algorithms $\mathcal{F}$). Therefore, it attains the instance-optimality over sequential decoding algorithms family $\mathcal{F}$.*

*Proof.* For any algorithm $\mathcal{A} \in \mathcal{F}$, its $b_n$ can be written as a function of the sequence $x_{1:n-1}, \tilde{x}_n$ with $0 \leqslant b_n(\tilde{x}_n, x_{1:n-1}) \leqslant 1$.[9] Based on this, we define the new function $\epsilon_n : \mathcal{V} \times \mathcal{V}^{n-1} \mapsto \mathbb{R}$ according to the following equation:

$$b_n(\tilde{x}_n, x_{1:n-1}) = \min \left\{ 1, \frac{q_n(\tilde{x}_n | x_{1:n-1}) + \epsilon_n(\tilde{x}_n, x_{1:n-1})}{p_n(\tilde{x}_n | x_{1:n-1})} \right\}. \tag{9}$$

Indeed, we can choose $\epsilon_n := b_n \cdot p_n - q_n$, and the validity of the definition is guaranteed by the Lemma 1. Recall $\mathcal{A} \in \mathcal{F}$ is a distribution unbiased algorithm w.r.t. $q$. Let $x_1, \ldots, x_n$ be the validated sequence of $\mathcal{A}$, then we have

$$\mathbb{P}_n^{\mathcal{A}}(x_n | x_{1:n-1}) = \frac{\mathbb{P}_n^{\mathcal{A}}(x_{1:n})}{\mathbb{P}_n^{\mathcal{A}}(x_{1:n-1})} = \frac{q_n(x_{1:n})}{q_n(x_{1:n-1})} = q_n(x_n | x_{1:n-1}). \tag{10}$$

On the other hand, let $\tilde{x}_n \sim p_n(\cdot | x_{1:n-1})$, the decomposition holds

$$\mathbb{P}_n^{\mathcal{A}}(x_n | x_{1:n-1}) = \mathbb{P}_n^{\mathcal{A}}(x_n, \tilde{x}_n \text{ accept} | x_{1:n-1}) + \mathbb{P}_n^{\mathcal{A}}(x_n, \tilde{x}_n \text{ reject} | x_{1:n-1})$$

$$= \mathbb{P}_n^{\mathcal{A}}(\tilde{x}_n \text{ accept} | \tilde{x}_n = x_n, x_{1:n-1}) \mathbb{P}_n^{\mathcal{A}}(\tilde{x}_n = x_n | x_{1:n-1})$$

$$+ \mathbb{P}_n^{\mathcal{A}}(x_n | \tilde{x}_n \text{ reject}, x_{1:n-1}) \mathbb{P}_n^{\mathcal{A}}(\tilde{x}_n \text{ reject} | x_{1:n-1})$$

$$= b_n(x_n, x_{1:n-1}) \cdot p_n(x_n | x_{1:n-1}) + \mathbb{P}_n^{\mathcal{A}}(x_n | \tilde{x}_n \text{ reject}, x_{1:n-1}) \cdot \mathbb{P}_n^{\mathcal{A}}(\tilde{x}_n \text{ reject} | x_{1:n-1})$$

$$= b_n(x_n, x_{1:n-1}) \cdot p_n(x_n | x_{1:n-1}) + \mathbb{P}_n^{\mathcal{A}}(x_n | \tilde{x}_n \text{ reject}, x_{1:n-1}) \cdot (1 - \sum_x b_n(x, x_{1:n-1}) p_n(x | x_{1:n-1})), \tag{11}$$

where the third equal sign uses $\mathbb{P}_n^{\mathcal{A}}(\tilde{x}_n = x_n | x_{1:n-1}) = p_n(x_n | x_{1:n-1})$ and the last equal sign is due to

$$\mathbb{P}_n^{\mathcal{A}}(\tilde{x}_n \text{ reject} | x_{1:n-1}) = 1 - \mathbb{P}_n^{\mathcal{A}}(\tilde{x}_n \text{ accept} | x_{1:n-1}) = 1 - \sum_x \mathbb{P}_n^{\mathcal{A}}(\tilde{x}_n \text{ accept}, \tilde{x}_n = x | x_{1:n-1})$$

$$= 1 - \sum_x \mathbb{P}_n^{\mathcal{A}}(\tilde{x}_n \text{ accept} | \tilde{x}_n = x, x_{1:n-1}) \mathbb{P}_n^{\mathcal{A}}(\tilde{x}_n = x | x_{1:n-1}) = 1 - \sum_x b_n(x, x_{1:n-1}) p_n(x | x_{1:n-1}).$$

Let's do some further simplifications. First off,

$$b_n(x_n, x_{1:n-1}) \cdot p_n(x_n | x_{1:n-1}) = p_n(x_n | x_{1:n-1}) \min \left\{ 1, \frac{q_n(x_n | x_{1:n-1}) + \epsilon_n(x_n, x_{1:n-1})}{p_n(x_n | x_{1:n-1})} \right\}$$

$$= \min \{ p_n(x_n | x_{1:n-1}), q_n(x_n | x_{1:n-1}) + \epsilon_n(x_n, x_{1:n-1}) \}, \tag{12}$$

and

$$1 - \sum_x b_n(x, x_{1:n-1}) p_n(x | x_{1:n-1}) = 1 - \sum_x \min \{ p_n(x | x_{1:n-1}), q_n(x_n | x_{1:n-1}) + \epsilon_n(x, x_{1:n-1}) \}$$

$$= \sum_x \max \{ 0, p_n(x | x_{1:n-1}) - q_n(x | x_{1:n-1}) - \epsilon_n(x, x_{1:n-1}) \} \tag{13}$$

---

[9]For Speculative Decoding, its $b_n(\tilde{x}_n, x_{1:n-1}) = \min \left\{ 1, \frac{q_n(\tilde{x}_n | x_{1:n-1})}{p_n(\tilde{x}_n | x_{1:n-1})} \right\}$.

Plug (12) and (13) into (11), and then plug (11) into (10) to obtain

$$q_n(x_n|x_{1:n-1}) = \min\{p_n(x_n|x_{1:n-1}), q_n(x_n|x_{1:n-1}) + \epsilon_n(x_n, x_{1:n-1})\}$$
$$+ \mathbb{P}_n^{\mathcal{A}}(x_n|\tilde{x}_n \text{ reject}, x_{1:n-1}) \cdot \sum_x (p_n(x|x_{1:n-1}) - q_n(x|x_{1:n-1}) - \epsilon_n(x, x_{1:n-1}))_+. \quad (14)$$

Now, we define the token set $\mathcal{V}_+ = \{x : \exists\, n, x_{1:n-1}\ s.t.\ \epsilon_n(x, x_{1:n-1}) > 0\}$ and similarly the token set $\mathcal{V}_- = \{x : \exists\, n, x_{1:n-1}\ s.t.\ \epsilon_n(x, x_{1:n-1}) \leqslant 0\}$. Next, we show $\mathcal{V}_+ = \varnothing$, and $\mathcal{V}_- = \mathcal{V}$.

**Case1.** If $p_n(x_n|x_{1:n-1}) \geqslant q_n(x_n|x_{1:n-1}) + \epsilon_n(x_n, x_{1:n-1})$, then by (14) we have

$$q_n(x_n|x_{1:n-1}) = q_n(x_n|x_{1:n-1}) + \epsilon_n(x_n, x_{1:n-1})$$
$$+ \mathbb{P}_n^{\mathcal{A}}(x_n|\tilde{x}_n \text{ reject}, \tilde{x}_n = x_n, x_{1:n-1}) \cdot \sum_x (p_n(x|x_{1:n-1}) - q_n(x|x_{1:n-1}) - \epsilon_n(x, x_{1:n-1}))_+$$

$$\geqslant q_n(x_n|x_{1:n-1}) + \epsilon_n(x_n, x_{1:n-1}),$$

and this implies $0 \geqslant \epsilon_n(x_n, x_{1:n-1})$, which means $x_n \in \mathcal{V}_-$;

**Case2.** If $p_n(x_n|x_{1:n-1}) < q_n(x_n|x_{1:n-1}) + \epsilon_n(x_n, x_{1:n-1})$, then by (14) we have

$$q_n(x_n|x_{1:n-1}) = p_n(x_n|x_{1:n-1})$$
$$+ \mathbb{P}_n^{\mathcal{A}}(x_n|\tilde{x}_n \text{ reject}, \tilde{x}_n = x_n, x_{1:n-1}) \cdot \sum_x (p_n(x|x_{1:n-1}) - q_n(x|x_{1:n-1}) - \epsilon_n(x, x_{1:n-1}))_+$$

$$\geqslant p_n(x_n|x_{1:n-1}),$$

and this implies

$$\epsilon_n(x_n, x_{1:n-1}) = b_n(x_n, x_{1:n-1})p_n(x_n|x_{1:n-1}) - q_n(x_n|x_{1:n-1})$$
$$\leqslant p_n(x_n|x_{1:n-1}) - q_n(x_n|x_{1:n-1}) \leqslant 0,$$

which means $x_n \in \mathcal{V}_-$.

Therefore, combining the two cases we always have $x \in \mathcal{V}_-$, which indicates $\mathcal{V}_+ = \varnothing$. By (9), this implies for all $n$, $b_n \leqslant \min\{1, \frac{q_n}{p_n}\}$. Finally, by Lemma 2, this implies $\mathbb{E}_{\mathcal{P}}^{\mathcal{A}}[N_{\text{rej}}] \geqslant \mathfrak{C}(\mathcal{P})$. Since $\mathcal{A} \in \mathcal{F}$ is arbitrary, this concludes the proof. $\qquad\square$

**Corollary 1.** *For any algorithm $\mathcal{A} \in \mathcal{F}$, it follows $\forall\, n \in [T], x \in \mathcal{V}$ and $x_{1:n-1}$, $b_n(x, x_{1:n-1}) \leqslant \min\left\{1, \frac{q_n(x|x_{1:n-1})}{p_n(x|x_{1:n-1})}\right\}$. In this case, the distribution $\mathcal{P}_n$ is defined as:*

$$\mathcal{P}_n(x|x_{1:n-1}) = \frac{q_n(x|x_{1:n-1}) - \min\{p_n(x|x_{1:n-1}), q_n(x|x_{1:n-1}) + \epsilon_n(x, x_{1:n-1})\}}{\sum_x (p_n(x|x_{1:n-1}) - q_n(x|x_{1:n-1}) - \epsilon_n(x|x_{1:n-1}))_+}.$$

*Applying $\epsilon_n = b_n p_n - q_n$, this is equivalent to*

$$\mathcal{P}_n(x|x_{1:n-1}) = \frac{q_n(x|x_{1:n-1}) - b_n(x, x_{1:n-1})p_n(x|x_{1:n-1})}{\sum_x (1 - b_n(x, x_{1:n-1}))p_n(x|x_{1:n-1})}.$$

*Proof of Corollary 1.* We reutilize (14) here and call it (15).

$$q_n(x_n|x_{1:n-1}) = \min\{p_n(x_n|x_{1:n-1}), q_n(x_n|x_{1:n-1}) + \epsilon_n(x_n, x_{1:n-1})\}$$
$$+ \mathbb{P}_n^{\mathcal{A}}(x_n|\tilde{x}_n \text{ reject}, x_{1:n-1}) \cdot \sum_x (p_n(x|x_{1:n-1}) - q_n(x|x_{1:n-1}) - \epsilon_n(x, x_{1:n-1}))_+. \quad (15)$$

By two cases discussion as in the proof of Theorem 8, we have $\forall\, \mathcal{A} \in \mathcal{F}$, it follows $\forall\, n \in [T], x \in \mathcal{V}$ and $x_{1:n-1}$, $b_n(x, x_{1:n-1}) \leqslant \min\left\{1, \frac{q_n(x|x_{1:n-1})}{p_n(x|x_{1:n-1})}\right\}$. Then we can directly solve (15) to obtain

$$\mathbb{P}_n^{\mathcal{A}}(x|\tilde{x}_n \text{ reject}, x_{1:n-1}) = \frac{q_n(x|x_{1:n-1}) - \min\{p_n(x|x_{1:n-1}), q_n(x|x_{1:n-1}) + \epsilon_n(x, x_{1:n-1})\}}{\sum_x (p_n(x|x_{1:n-1}) - q_n(x|x_{1:n-1}) - \epsilon_n(x|x_{1:n-1}))_+}.$$

Lastly, we verify such a $\mathbb{P}_n^{\mathcal{A}}(x_n|\tilde{x}_n \text{ reject}, x_{1:n-1})$ is a valid distribution. First of all, since $\epsilon_n = b_n \cdot p_n - q_n$, then $b_n(x, x_{1:n-1}) \leqslant \min\left\{1, \frac{q_n(x|x_{1:n-1})}{p_n(x|x_{1:n-1})}\right\}$ implies $\epsilon_n(x, x_{1:n-1}) \leqslant 0$, and this further implies

$$q_n(x|x_{1:n-1}) - \min\{p_n(x|x_{1:n-1}), q_n(x|x_{1:n-1}) + \epsilon_n(x, x_{1:n-1})\}$$
$$\geqslant q_n(x|x_{1:n-1}) - \min\{p_n(x|x_{1:n-1}), q_n(x|x_{1:n-1})\} \geqslant 0$$

which implies $\mathbb{P}_n^{\mathcal{A}}(x_n|\tilde{x}_n \text{ reject}, x_{1:n-1}) \geqslant 0$. Second,

$$\sum_x \mathbb{P}_n^{\mathcal{A}}(x|\tilde{x}_n \text{ reject}, x_{1:n-1}) = \sum_x \frac{q_n(x|x_{1:n-1}) - \min\{p_n(x|x_{1:n-1}), q_n(x|x_{1:n-1}) + \epsilon_n(x, x_{1:n-1})\}}{\sum_x(p_n(x|x_{1:n-1}) - q_n(x|x_{1:n-1}) - \epsilon_n(x|x_{1:n-1}))_+}$$

$$= \frac{1 - \sum_x \min\{p_n(x|x_{1:n-1}), q_n(x|x_{1:n-1}) + \epsilon_n(x, x_{1:n-1})\}}{\sum_x(p_n(x|x_{1:n-1}) - q_n(x|x_{1:n-1}) - \epsilon_n(x|x_{1:n-1}))_+}$$

$$= \frac{1 + \sum_x \max\{-p_n(x|x_{1:n-1}), -q_n(x|x_{1:n-1}) - \epsilon_n(x, x_{1:n-1})\}}{\sum_x(p_n(x|x_{1:n-1}) - q_n(x|x_{1:n-1}) - \epsilon_n(x|x_{1:n-1}))_+}$$

$$= \frac{\sum_x p_n(x|x_{1:n-1}) + \sum_x \max\{-p_n(x|x_{1:n-1}), -q_n(x|x_{1:n-1}) - \epsilon_n(x, x_{1:n-1})\}}{\sum_x(p_n(x|x_{1:n-1}) - q_n(x|x_{1:n-1}) - \epsilon_n(x|x_{1:n-1}))_+}$$

$$= \frac{\sum_x \max\{0, p_n(x|x_{1:n-1}) - q_n(x|x_{1:n-1}) - \epsilon_n(x, x_{1:n-1})\}}{\sum_x(p_n(x|x_{1:n-1}) - q_n(x|x_{1:n-1}) - \epsilon_n(x|x_{1:n-1}))_+} = 1.$$

This concludes the proof. $\qquad\square$

**Lemma 1.** *For any $0 \leqslant b_n(\tilde{x}_n, x_{1:n-1}) \leqslant 1$, there exists $\epsilon_n(\tilde{x}_n, x_{1:n-1}) \in \mathbb{R}$ such that (9) holds true.*

*Proof of Lemma 1.* Indeed, set $\epsilon_n = b_n \cdot p_n - q_n$, then

$$\min\left\{1, \frac{q_n(\tilde{x}_n|x_{1:n-1}) + \epsilon_n(\tilde{x}_n, x_{1:n-1})}{p_n(\tilde{x}_n|x_{1:n-1})}\right\} = \min\{1, b_n(\tilde{x}_n, x_{1:n-1})\} = b_n(\tilde{x}_n, x_{1:n-1})$$

where the first equal sign uses that $\tilde{x}_n$ is sampled from $p_n(\cdot|x_{1:n-1})$ so $p_n(\tilde{x}_n|x_{1:n-1}) > 0$, and the second equal sign uses $0 \leqslant b_n \leqslant 1$. $\qquad\square$

**Lemma 2.** *For any instance $\mathcal{P} = (p, q)$, let $\mathcal{F} := \{\mathcal{A} : \mathcal{A} \text{ is a realization of Algorithm 2 s.t. } \mathbb{P}_t^{\mathcal{A}} = q_t \ \forall t \ (\text{i.e., unbiased})\}$. Suppose there is an $\mathcal{A} \in \mathcal{F}$ such that $\mathbb{E}_{\mathcal{P}}^{\mathcal{A}}[N_{rej}] < \mathfrak{C}(\mathcal{P})$, then there exists a $b_n$ in Line 8 of Template 2 such that $\exists x, x_{1:n-1}$*

$$b_n(x|x_{1:n-1}) > \min\left\{1, \frac{q_n(x|x_{1:n-1})}{p_n(x|x_{1:n-1})}\right\}.$$

*Proof of Lemma 2.* Suppose for all $n, x, x_{1:n-1}, b_n(x|x_{1:n-1}) \leqslant \min\left\{1, \frac{q_n(x|x_{1:n-1})}{p_n(x|x_{1:n-1})}\right\}$. We define random variables $R_n \in \{0, 1\}$ indicating whether the $n$-th token is rejected (1 is rejected). Therefore, the expected number of rejection is

$$\mathbb{E}^{\mathcal{A}}\left[\sum_{n=1}^T R_n\right] = \sum_{n=1}^T \mathbb{E}^{\mathcal{A}}[R_n] = \sum_{n=1}^T \mathbb{E}_{x_{1:n-1}\sim\mathbb{P}^{\mathcal{A}}}\left[\mathbb{E}^{\mathcal{A}}[R_n|x_{1:n-1}]\right] = \sum_{n=1}^T \mathbb{E}_{x_{1:n-1}\sim q}\left[\mathbb{E}^{\mathcal{A}}[R_n|x_{1:n-1}]\right].$$

Here the second to last equality comes from the tower property and the last equal signs is due to $\mathcal{A}$ is unbiased. Next, denote $\tilde{x}_n \sim p_n(\cdot|x_{1:n-1})$ to be the candidate token, we have

$$\mathbb{E}^{\mathcal{A}}[R_n|x_{1:n-1}] = \mathbb{P}^{\mathcal{A}}[R_n = 1|x_{1:n-1}] = \sum_{\tilde{x}_n} \mathbb{P}^{\mathcal{A}}[R_n = 1, \tilde{x}_n|x_{1:n-1}]$$

$$= \sum_{\tilde{x}_n} \mathbb{P}^{\mathcal{A}}[R_n = 1|\tilde{x}_n, x_{1:n-1}]\mathbb{P}^{\mathcal{A}}[\tilde{x}_n|x_{1:n-1}] = \sum_{\tilde{x}_n} \mathbb{P}^{\mathcal{A}}[R_n = 1|\tilde{x}_n, x_{1:n-1}]p_n[\tilde{x}_n|x_{1:n-1}]$$

$$= \sum_{\tilde{x}_n}(1 - b_n) \cdot p_n[\tilde{x}_n|x_{1:n-1}] \geqslant \sum_{\tilde{x}_n}(1 - \min\left\{1, \frac{q_n(\tilde{x}_n|x_{1:n-1})}{p_n(\tilde{x}_n|x_{1:n-1})}\right\}) \cdot p_n[\tilde{x}_n|x_{1:n-1}]$$

$$= \sum_{\tilde{x}_n}[p_n(\tilde{x}_n|x_{1:n-1}) - q_n(\tilde{x}_n|x_{1:n-1})]_+ = \mathsf{TV}[(p_n(\cdot|x_{1:n-1}) - q_n(\cdot|x_{1:n-1})]$$

and this implies

$$\mathbb{E}_{\mathcal{P}}^{\mathcal{A}}[N_{\text{rej}}] = \mathbb{E}^{\mathcal{A}}\left[\sum_{n=1}^T r_n\right] \geqslant \sum_{n=1}^T \mathbb{E}_{x_{1:n-1}\sim q}[\mathsf{TV}[(p_n(\cdot|x_{1:n-1}) - q_n(\cdot|x_{1:n-1})]] = \mathfrak{C}(\mathcal{P})$$

contradicts $\mathbb{E}^{\mathcal{A}}_{\mathcal{P}}[N_{\mathrm{rej}}] < \mathfrak{C}(\mathcal{P})$! This concludes the proof.

$\square$

---

**Algorithm 4** Batch Speculative Sampling

---

1: **Init**: Horizon $T$, Distributions $q_t$ and $p_t$, with $q = \mathbb{P}^{\text{LLM}}$, $p = \mathbb{P}^{\text{Draft}}$. Lookahead $K = \infty$.
2: **while** $n < T$ **do**
3:     Reset $q_t = \mathbb{P}_t^{\text{LLM}}$ $\forall t$. $n_0 = n$.
4:     **for** $m = 1 : M$ $\diamond$Sample $M$ draft responses in parallel$\diamond$ **do**
5:       **for** $t = n : T$ **do**
6:         Sample $\tilde{x}_t^m \sim p_t(\cdot|x_{1:n-1}, \tilde{x}_{n:t-1}^m)$.
7:       **end for**
8:     **end for**
9:     Obtain logits $q_n(\cdot|x_{1:n-1})$, $\ldots$, $q_T(\cdot|x_{1:n-1}, \tilde{x}_{n:T}^m)$, $\forall m \in [M]$ in parallel for $\tilde{x}_{n:T}^m$.
10:     ——— $\diamond$ Verification Begins ———
11:     Set Sample=False.
12:     **for** $m = 1, \ldots, M$ **do**
13:       **for** $t = n : T$ **do**
14:         Sample $r \sim \text{Uniform}[0, 1]$.
15:         **if** $r \leqslant \min\left\{1, \frac{q_t(\tilde{x}_t^m|x_{1:n-1}, \tilde{x}_{n:t-1}^m)}{p_t(\tilde{x}_t^m|x_{1:n-1}, \tilde{x}_{n:t-1}^m)}\right\}$ **then**
16:           Accept with $x_n = \tilde{x}_t^m$. $n \leftarrow n + 1$. Sample=True.
17:         **else**
18:           **if** $t = n_0$ **then**
19:             Update $q_n \leftarrow \left[q_n(\cdot|x_{1:n-1}) - p_n(\cdot|x_{1:n-1})\right]_+$. Break. //Here $n_0$ equals $n$.
20:           **else**
21:             $\diamond$ Rejection Sample $x_n \sim \left[q_n(\cdot|x_{1:n-1}) - p_n(\cdot|x_{1:n-1})\right]_+$. $n \leftarrow n + 1$. Break.
22:           **end if**
23:         **end if**
24:       **end for**
25:       **if** Sample= TRUE **then**
26:         Break.
27:       **else**
28:         $\diamond$ Rejection Sample $x_n \sim \left[q_n(\cdot|x_{1:n-1}) - p_n(\cdot|x_{1:n-1})\right]_+$. $n \leftarrow n + 1$. Break.
29:       **end if**
30:     **end for**
31: **end while**

---

# D  Batch Speculative Decoding

We split the proofs for unbiasedness and rejections in two parts.

**Theorem 9** (Unbiasedness of Theorem 3)**.** *Denote the Algorithm 4 as $\mathcal{A}_\mathcal{B}$. For any sequence $x_{1:T}$ (where $x_i \in \mathcal{V}$), we have*

$$\mathbb{P}_T^{\mathcal{A}_\mathcal{B}}(x_1, \ldots, x_T) = \mathbb{P}_T^{LLM}(x_1, \ldots, x_T).$$

*Proof.* **Step1:** Let $x_{1:n-1}$ be the accepted tokens up to $n - 1$. We first show $\mathbb{P}_n^{\mathcal{A}_\mathcal{B}}(x_n|x_{1:n-1}) = \mathbb{P}_n^{LLM}(x_n|x_{1:n-1}) \; \forall x_n \in \mathcal{V}$.

We partition the generation of $x_n$ into two cases: (i). accepted as the first token of the $m$-th responses ($m = 1, \ldots, M$), or rejected by all $M$ responses and sampled by Line 28 of Algorithm 4; (ii). accepted/rejected as the $t$-th token of the $m$-th responses ($m = 1, \ldots, M$) for $t \geqslant 2$.

**For the second case.** Similar to the standard speculative decoding, let $\tilde{x}_n^m \sim p_n(\cdot|x_{1:n-1})$, then

$$\mathbb{P}_n^{\mathcal{A}_\mathcal{B}}(x_n|x_{1:n-1}) = \mathbb{P}_n^{\mathcal{A}_\mathcal{B}}(x_n, \tilde{x}_n^m \text{ acc}|x_{1:n-1}) + \mathbb{P}_n^{\mathcal{A}_\mathcal{B}}(x_n, \tilde{x}_n^m \text{ rej}|x_{1:n-1})$$

$$= \mathbb{P}_n^{\mathcal{A}_\mathcal{B}}(\tilde{x}_n^m = x_n|x_{1:n-1})\mathbb{P}_n^{\mathcal{A}_\mathcal{B}}(\tilde{x}_n^m \text{ acc}|\tilde{x}_n^m = x_n, x_{1:n-1})$$

$$+ \mathbb{P}_n^{\mathcal{A}_\mathcal{B}}(\tilde{x}_n^m \text{ rej}|x_{1:n-1})\mathbb{P}_n^{\mathcal{A}_\mathcal{B}}(x_n|\tilde{x}_n^m \text{ rej}, x_{1:n-1})$$

$$= p_n(x_n|x_{1:n-1}) \min\{1, \frac{q_n(x_n|x_{1:n-1})}{p_n(x_n|x_{1:n-1})}\}$$

$$+ \sum_{x'} \max\{0, q_n(x'|x_{1:n-1}) - p_n(x'|x_{1:n-1})\}\frac{\max\{0, q_n(x_n|x_{1:n-1}) - p_n(x_n|x_{1:n-1})\}}{\sum_{x'} \max\{0, q_n(x'|x_{1:n-1}) - p_n(x'|x_{1:n-1})\}} = q_n(x_n|x_{1:n-1}).$$

By the construction of Algorithm 4 (Line 3), when $x_n$ is accepted/rejected as the $t(\geqslant 2)$-th draft token of certain response, we have $q_n(x_n|x_{1:n-1}) = \mathbb{P}_n^{LLM}(x_n|x_{1:n-1})$. This gives $\mathbb{P}_n^{\mathcal{A}_\mathcal{B}}(x_n|x_{1:n-1}) = \mathbb{P}_n^{LLM}(x_n|x_{1:n-1})$.

**For the first case.** This part of the proof largely follows Theorem 4.2 of [26]. In this case, the $n$-th generated token $x_n$ has $M + 1$ possibilities: accepted at the $m$-th response or rejected by all $M$ responses and sample at Line 28. Since the algorithm will iterate all $M$ responses if not accepted, we denote $q_n^m$ as the $q_n$ for the $m$-th response. Then we have the recursion

$$q_n^{m+1} = \frac{\max\{0, q_n^m - p_n\}}{r_m},$$

where $q_n^1 = \mathbb{P}_n^{LLM}$ and $r_m$ is the rejection probability satisfies

$$r_m = 1 - \mathbb{P}_n^{\mathcal{A}_\mathcal{B}}(\tilde{x}_n^m \text{ acc}|x_{1:n-1}) = 1 - \sum_x \mathbb{P}_n^{\mathcal{A}_\mathcal{B}}(\tilde{x}_n^m = x|x_{1:n-1})\mathbb{P}_n^{\mathcal{A}_\mathcal{B}}(\tilde{x}_n^m \text{ acc}|\tilde{x}_n^m = x, x_{1:n-1})$$

$$= 1 - \sum_x p_n(x|x_{1:n-1}) \min\{1, \frac{q_n^m(x|x_{1:n-1})}{p_n(x|x_{1:n-1})}\} = \sum_x \max\{0, q_n^m(x|x_{1:n-1}) - p_n(x|x_{1:n-1})\}.$$

Denote $E_m = \{m\text{-th response rejected}\}$ and $E_{1:m} = \{1 : m\text{-th responses all rejected}\}$, then

$$\mathbb{P}_n^{\mathcal{A}_\mathcal{B}}(x_n|x_{1:n-1}) = \mathbb{P}_n^{\mathcal{A}_\mathcal{B}}(x_n, E_1^c|x_{1:n-1}) + \mathbb{P}_n^{\mathcal{A}_\mathcal{B}}(x_n, E_1|x_{1:n-1})$$

$$= \min\{p_n(x_n|x_{1:n-1}), q_{n|}^1(x_n|x_{1:n-1})\} + \mathbb{P}_n^{\mathcal{A}_\mathcal{B}}(x_n, E_1|x_{1:n-1})$$

$$= \min\{p_n(x_n|x_{1:n-1}), q_n^1(x_n|x_{1:n-1})\} + r_1 \cdot \mathbb{P}_n^{\mathcal{A}_\mathcal{B}}(x_n|E_1, x_{1:n-1})$$

Denote $\mathbb{P}_n^{\mathcal{A}_\mathcal{B}}(x_n|x_{1:n-1}) := \mathbb{P}_n^{\mathcal{A}_\mathcal{B}}(x_n|E_0, x_{1:n-1})$, by similar calculation we have in general

$$\mathbb{P}_n^{\mathcal{A}_\mathcal{B}}(x_n|E_{0:m-1}, x_{1:n-1}) = \min\{p_n(x_n|x_{1:n-1}), q_{n|}^m(x_n|x_{1:n-1})\} + r_m \cdot \mathbb{P}_n^{\mathcal{A}_\mathcal{B}}(x_n|E_{0:m}, x_{1:n-1}).$$

Next we prove $\mathbb{P}_n^{\mathcal{A}_\mathcal{B}}(\cdot|E_{0:m}, x_{1:n-1}) = q_n^{m+1}(\cdot|x_{1:n-1}) \; \forall m \in \{0, 1, \ldots, M\}$ by backward induction. First of all, $\mathbb{P}_n^{\mathcal{A}_\mathcal{B}}(\cdot|E_{0:M}, x_{1:n-1}) = \left[q_{n+1}^M(\cdot|x_{1:n-1}) - p_{n+1}(\cdot|x_{1:n-1})\right]_+ = \frac{\max\{0, q_n^M - p_n\}(\cdot|x_{1:n-1})}{r_M} = q_n^{M+1}$. Suppose $\mathbb{P}_n^{\mathcal{A}_\mathcal{B}}(\cdot|E_{0:m+1}, x_{1:n-1}) = q_n^{m+2}(\cdot|x_{1:n-1})$, then

$$\mathbb{P}_n^{\mathcal{A}_\mathcal{B}}(\cdot|E_{0:m}, x_{1:n-1}) = \mathbb{P}_n^{\mathcal{A}_\mathcal{B}}(\cdot, E_{m+1}^c|E_{0:m}, x_{1:n-1}) + \mathbb{P}_n^{\mathcal{A}_\mathcal{B}}(\cdot, E_{m+1}|E_{0:m}, x_{1:n-1})$$

$$= \min\{p_n(\cdot|x_{1:n-1}), q_n^{m+1}(\cdot|x_{1:n-1})\} + \mathbb{P}_n^{\mathcal{A}_\mathcal{B}}(\cdot, E_{m+1}|E_{0:m}, x_{1:n-1})$$

$$= \min\{p_n(\cdot|x_{1:n-1}), q_n^{m+1}(\cdot|x_{1:n-1})\} + r_{m+1} \cdot \mathbb{P}_n^{\mathcal{A}_\mathcal{B}}(\cdot|E_{0:m+1}, x_{1:n-1})$$

$$= \min\{p_n(\cdot|x_{1:n-1}), q_n^{m+1}(\cdot|x_{1:n-1})\} + \max\{0, q_n^{m+1} - p_n\} \equiv q_n^{m+1}.$$

In particular, we take $m = 0$ to obtain

$$\mathbb{P}_n^{\mathcal{A}_{\mathcal{B}}}(\cdot|x_{1:n-1}) = q_n^1(\cdot|x_{1:n-1}) = \mathbb{P}_n^{\text{LLM}}(\cdot|x_{1:n-1}).$$

Combining both cases we finish the proof of Step1.

**Step2:** For any $n$, first of all we have $\mathbb{P}^{\mathcal{A}_{\mathcal{B}}}(x_0) = \mathbb{P}^{\text{LLM}}(x_0)$ since $x_0$ is the prompt. Suppose $\mathbb{P}_{n-1}^{\mathcal{A}_{\mathcal{B}}}(x_{1:n-1}) = \mathbb{P}_{n-1}^{\text{LLM}}(x_{1:n-1})$, $\forall x_{1:n-1}$, then by Step1

$$\mathbb{P}_n^{\mathcal{A}_{\mathcal{B}}}(x_{1:n}) = \mathbb{P}_n^{\mathcal{A}_{\mathcal{B}}}(x_n|x_{1:n-1})\mathbb{P}_{n-1}^{\mathcal{A}_{\mathcal{B}}}(x_{1:n-1}) = \mathbb{P}_n^{\mathcal{A}_{\mathcal{B}}}(x_n|x_{1:n-1})\mathbb{P}_{n-1}^{\text{LLM}}(x_{1:n-1}) = \mathbb{P}_n^{\text{LLM}}(x_{1:n})$$

where the second equal sign is by induction and the third equal sign is by Step1. This finish the proof.

$\square$

## D.1 Expected Rejections for Batch Speculative Decoding (Proof for the second part of Theorem 3)

In this section, we derive the number of expected rejections using the batch speculative decoding. However, the analysis is more involved than the Algorithm 4 due to the parallel response structure, since, given the verified token $x_{1:n-1}$, the probability of $n$-th token being rejected does not possess a unified expression and depends on the location of $x_{n-1}$. We detail this below.

We recall the notion in Theorem 1 that random variables $R_n \in \{0, 1\}$ indicates whether the $n$-th token is rejected (1 is rejected). Therefore, the expected number of rejection is

$$\mathbb{E}\left[\sum_{n=1}^T R_n\right] = \sum_{n=1}^T \mathbb{E}[R_n] = \sum_{n=1}^T \mathbb{E}\left[\mathbb{E}[R_n|x_{1:n-1}]\right], \tag{16}$$

where the last equality comes from the tower property of expectation and we assume $x_0$ is a given initial token and $x_{1:0} = \{x_0\}$. Then

$$\mathbb{E}[R_n|x_{1:n-1}] = \mathbb{P}^{\mathcal{A}}(n\text{-th token rej}|x_{1:n-1}) = 1 - \mathbb{P}^{\mathcal{A}}(n\text{-th token acc}|x_{1:n-1}).$$

In this scenario, we cannot obtain $\mathbb{P}^{\mathcal{A}}(n\text{-th token rej}|x_{1:n-1}) = \text{TV}(p_n(\cdot|x_{1:n-1}), \mathbb{P}_n^{\text{LLM}}(\cdot|x_{1:n-1}))$ since the conditional rejection probability implicitly encodes the location of $(n-1)$-th token: whether $\tilde{x}_{n-1} \sim p(\cdot|x_{1:n-1})$ is rejected (at the root of the tree) or $\tilde{x}_{n-1} \sim p(\cdot|x_{1:n-1})$ is accepted (at the branch of the tree). To formalize this, given validated token $x_{1:n-1}$, we denote $q_n^m(\cdot|x_{1:n-1})$ to be the $m$-th rejection distribution, then by the construction of Algorithm 4 (Line 19),

$$q_n^{m+1} = \frac{\max\{0, q_n^m - p_n\}}{r_m}, \quad \forall m \in [M].$$

Here $r_m = \sum_{x'} \max\{0, q_n^m(x'|x_{1:n-1}) - p_n(x'|x_{1:n-1})\}$ is normalizing factor and $q_n^1 = \mathbb{P}^{\text{LLM}}$. Let $\tilde{x}_n \sim p(\cdot|x_{1:n-1})$, then $\mathbb{P}^{\mathcal{A}}(n\text{-th token acc}|x_{1:n-1}) = \mathbb{P}^{\mathcal{A}}(\tilde{x}_n \text{ acc}|x_{1:n-1})$. Next, we compute the quantity $\mathbb{P}^{\mathcal{A}}(\tilde{x}_n \text{ acc}|x_{1:n-1})$.

We begin by first considering $\mathbb{P}^{\mathcal{A}}(\tilde{x}_n \text{ acc}, \tilde{x}_n = x_n|x_{1:n-1})$. Let $\tilde{x}_{n-1} \sim p(\cdot|x_{1:n-2})$, then there are two cases: $\tilde{x}_{n-1}$ accepted or rejected. We have

$$\mathbb{P}^{\mathcal{A}}(\tilde{x}_n \text{ acc}, \tilde{x}_n = x_n|x_{1:n-1}) = \mathbb{P}^{\mathcal{A}}(\tilde{x}_n \text{ acc}, \tilde{x}_n = x_n, \tilde{x}_{n-1}\text{acc}|x_{1:n-1}) + \mathbb{P}^{\mathcal{A}}(\tilde{x}_n \text{ acc}, \tilde{x}_n = x_n, \tilde{x}_{n-1}\text{rej}|x_{1:n-1})$$

$$= \underbrace{\mathbb{P}^{\mathcal{A}}(\tilde{x}_n \text{ acc}, \tilde{x}_n = x_n|\tilde{x}_{n-1}\text{acc}, x_{1:n-1})}_{p_a} \mathbb{P}^{\mathcal{A}}(\tilde{x}_{n-1}\text{acc}|x_{1:n-1})$$

$$+ \underbrace{\mathbb{P}^{\mathcal{A}}(\tilde{x}_n \text{ acc}, \tilde{x}_n = x_n|\tilde{x}_{n-1}\text{rej}, x_{1:n-1})}_{p_b} \mathbb{P}^{\mathcal{A}}(\tilde{x}_{n-1}\text{rej}|x_{1:n-1})$$

$$\tag{17}$$

**For $p_a$.** Given that $\tilde{x}_{n-1}$ is accepted, $\tilde{x}_n = x_n$ can only be accepted as the $t$-th token within the certain response for $t \geqslant 2$. This is because $\tilde{x}_{n-1}$ is accepted and has to be at least the first token in the response. In this case, $q_n(\cdot|x_{1:n-1}) = \mathbb{P}_n^{\text{LLM}}(\cdot|x_{1:n-1}) = q_n^1(\cdot|x_{1:n-1})$, then

$$p_a = \mathbb{P}^{\mathcal{A}}(\tilde{x}_n \text{ acc}|\tilde{x}_n = x_n, \tilde{x}_{n-1}\text{acc}, x_{1:n-1})\mathbb{P}^{\mathcal{A}}(\tilde{x}_n = x_n|\tilde{x}_{n-1}\text{acc}, x_{1:n-1})$$

$$= \min\{1, \frac{q_n(x_n|x_{1:n-1})}{p_n(x_n|x_{1:n-1})}\} \cdot p_n(x_n|x_{1:n-1}) = \min\{p_n(x_n|x_{1:n-1}), q_n(x_n|x_{1:n-1})\} \tag{18}$$

$$= \min\{p_n(x_n|x_{1:n-1}), q_n^1(x_n|x_{1:n-1})\}.$$

**For $p_b$.** Given that $\tilde{x}_{n-1}$ is rejected, $\tilde{x}_n = x_n$ can only be accepted as the first token within the certain response. This is because $\tilde{x}_{n-1}$ is rejected will restart the parallel tree. Then

$$
\begin{aligned}
p_b &= \mathbb{P}^{\mathcal{A}}(x_n|\tilde{x}_{n-1}\text{rej}, x_{1:n-1}) - \mathbb{P}^{\mathcal{A}}(\tilde{x}_n \text{ rej}, x_n|\tilde{x}_{n-1}\text{rej}, x_{1:n-1}) \\
&= \mathbb{P}_n^{\text{LLM}}(x_n|x_{1:n-1}) - \mathbb{P}^{\mathcal{A}}(\tilde{x}_n \text{ rej}, x_n|\tilde{x}_{n-1}\text{rej}, x_{1:n-1}) \\
&= \mathbb{P}_n^{\text{LLM}}(x_n|x_{1:n-1}) - \left(\prod_{m=1}^{M} r_m\right) q_n^{M+1}(x_n|x_{1:n-1}),
\end{aligned}
\tag{19}
$$

where the first equal sign comes from: since $\tilde{x}_{n-1}$rej implies $x_n$ represents the first token of the parallel tree, then it is identical to the proof of the first case of Step1 in Theorem 9. The second equal sign is from: $\tilde{x}_n$ is rejected means all $M$ responses since $x_n$ is the first token of the tree. The conditional rejection probability

$$
\begin{aligned}
&\mathbb{P}(\tilde{x}_n^m \text{ rej}|\tilde{x}_n^{1:m-1} \text{ rej}, x_{1:n-1}) = 1 - \mathbb{P}(\tilde{x}_n^m \text{ acc}|\tilde{x}_n^{1:m-1} \text{ rej}, x_{1:n-1}) \\
&= 1 - \sum_x \mathbb{P}_n^{\mathcal{A_B}}(\tilde{x}_n^m = x|\tilde{x}_n^{1:m-1} \text{ rej}, x_{1:n-1})\mathbb{P}_n^{\mathcal{A_B}}(\tilde{x}_n^m \text{ acc}|\tilde{x}_n^{1:m-1} \text{ rej}, \tilde{x}_n^m = x, x_{1:n-1}) \\
&= 1 - \sum_x p_n(x|x_{1:n-1})\min\{1, \frac{q_n^m(x|x_{1:n-1})}{p_n(x|x_{1:n-1})}\} = \sum_x \max\{0, q_n^m(x|x_{1:n-1}) - p_n(x|x_{1:n-1})\} = r_m,
\end{aligned}
$$

so by chain rule, the total rejection probability is $\prod_{m=1}^{M} r_m$.

Plug (18), (19) into (17) to obtain

$$
\begin{aligned}
\mathbb{P}^{\mathcal{A}}(\tilde{x}_n \text{ acc}, x_n|x_{1:n-1}) &= \min\{p_n(x_n|x_{1:n-1}), q_n^1(x_n|x_{1:n-1})\}\mathbb{P}^{\mathcal{A}}(\tilde{x}_{n-1}\text{acc}|x_{1:n-1}) \\
&+ [q_n^1(x_n|x_{1:n-1}) - (\prod_{m=1}^{M} r_m) \cdot q_n^{M+1}(x_n|x_{1:n-1})]\mathbb{P}^{\mathcal{A}}(\tilde{x}_{n-1}\text{rej}|x_{1:n-1})
\end{aligned}
\tag{20}
$$

which is equivalent to

$$
\begin{aligned}
&q_n^1(x_n|x_{1:n-1}) - \mathbb{P}^{\mathcal{A}}(\tilde{x}_n \text{ rej}, x_n|x_{1:n-1}) = \min\{p_n(x_n|x_{1:n-1}), q_n^1(x_n|x_{1:n-1})\}[1 - \mathbb{P}^{\mathcal{A}}(\tilde{x}_{n-1}\text{rej}|x_{1:n-1})] \\
&+ [q_n^1(x_n|x_{1:n-1}) - (\prod_{m=1}^{M} r_m) \cdot q_n^{M+1}(x_n|x_{1:n-1})]\mathbb{P}^{\mathcal{A}}(\tilde{x}_{n-1}\text{rej}|x_{1:n-1}) \\
&\Leftrightarrow \mathbb{P}^{\mathcal{A}}(\tilde{x}_n \text{ rej}, x_n|x_{1:n-1}) = \max\{0, q_n^1(x_n|x_{1:n-1}) - p_{n|}(x_n|x_{1:n-1}))\} \\
&- \left(\max\{0, q_n^1(x_n|x_{1:n-1}) - p_{n|}(x_n|x_{1:n-1}))\} - (\prod_{m=1}^{M} r_m)q_n^{M+1}(x_n|x_{1:n-1})\right)\mathbb{P}^{\mathcal{A}}(\tilde{x}_{n-1}\text{rej}|x_{1:n-1}) \\
&\Leftrightarrow \mathbb{P}^{\mathcal{A}}(\tilde{x}_n \text{ rej}, x_n|x_{1:n-1})\mathbb{P}^{\mathcal{A}}(x_{1:n-1}) = \max\{0, q_n^1(x_n|x_{1:n-1}) - p_n(x_n|x_{1:n-1}))\}\mathbb{P}^{\mathcal{A}}(x_{1:n-1}) \\
&- \left(\max\{0, q_n^1(x_n|x_{1:n-1}) - p_n(x_n|x_{1:n-1}))\} - (\prod_{m=1}^{M} r_m)q_n^{M+1}(x_n|x_{1:n-1})\right)\mathbb{P}^{\mathcal{A}}(\tilde{x}_{n-1}\text{rej}, x_{n-1}|x_{1:n-2})\mathbb{P}^{\mathcal{A}}(x_{1:n-2}) \\
&\Leftrightarrow \mathbb{P}^{\mathcal{A}}(\tilde{x}_n \text{ rej}, x_n|x_{1:n-1})q_n^1(x_{1:n-1}) = \max\{0, q_n^1(x_n|x_{1:n-1}) - p_n(x_n|x_{1:n-1}))\}q_n^1(x_{1:n-1}) \\
&- \left(\max\{0, q_n^1(x_n|x_{1:n-1}) - p_n(x_n|x_{1:n-1}))\} - (\prod_{m=1}^{M} r_m)q_n^{M+1}(x_n|x_{1:n-1})\right)\mathbb{P}^{\mathcal{A}}(\tilde{x}_{n-1}\text{rej}, x_{n-1}|x_{1:n-2})q_n^1(x_{1:n-2}),
\end{aligned}
\tag{21}
$$

where the first line uses $\mathbb{P}^{\mathcal{A}}(x_n|x_{1:n-1}) = q_n^1(x_n|x_{1:n-1})$, the second equivalence uses Bayes rule, the third equivalence uses $\mathbb{P}^{\mathcal{A}}(x_{1:n}) = q^1(x_{1:n}) = q_n^1(x_{1:n})$ again. Now denote $f(x_{1:n}) := \mathbb{P}^{\mathcal{A}}(\tilde{x}_n \text{ rej}, x_n|x_{1:n-1})q_n^1(x_{1:n-1})$, and sum over $x_{1:n}$ for the above to obtain

$$
\begin{aligned}
&\Leftrightarrow \mathbb{E}_{x_{1:n-1}\sim q^1}[\mathbb{P}^{\mathcal{A}}(\tilde{x}_n \text{ rej}|x_{1:n-1})] = \mathbb{E}_{x_{1:n-1}\sim q^1}[\mathsf{TV}(q^1(\cdot|x_{1:n-1}, p(\cdot|x_{1:n-1})))] \\
&- \bar{\mathbb{E}}_{x_{1:n-1}\sim f}\left[\mathsf{TV}(q^1, p)(x_{1:n-1}) - [\prod_{m=1}^{M} \mathsf{TV}(q^m, p)(x_{1:n-1})]\right],
\end{aligned}
$$

where by (21) pseudo-measure $f$ satisfies $\forall x_{1:n}$

$$f(x_{1:n}) = \max\{0, q_n^1(x_n|x_{1:n-1}) - p_n(x_n|x_{1:n-1}))\}q_n^1(x_{1:n-1})$$

$$- \left(\max\{0, q_n^1(x_n|x_{1:n-1}) - p_n(x_n|x_{1:n-1}))\} - (\prod_{m=1}^{M} r_m)q_n^{M+1}(x_n|x_{1:n-1})\right) f(x_{1:n-1}). \tag{22}$$

Here we used $r_m = \sum_x \max\{0, q_n^m(x|x_{1:n-1}) - p_n(x|x_{1:n-1})\} = \mathsf{TV}(q^m, p)(x_{1:n-1})$.

Plug the above back to (16), we finally have

$$\mathbb{E}^{\mathcal{A}_\mathcal{B}}[\sum_{n=1}^{T} R_n]$$

$$= \sum_{n=1}^{T} \mathbb{E}_{x_{1:n-1}\sim q^1}[\mathsf{TV}(q^1(\cdot|x_{1:n-1}, p(\cdot|x_{1:n-1})))]$$

$$- \sum_{n=1}^{T} \bar{\mathbb{E}}_{x_{1:n-1}\sim f}\left[\mathsf{TV}(q^1, p)(x_{1:n-1}) - [\prod_{m=1}^{M} \mathsf{TV}(q^m, p)(x_{1:n-1})]\right].$$

The formulation for $f$ is iteratively obtained in (22).

## D.2 Increasing batch size to inf doesn't help

**Proposition 1.** *Let $f^M$ be the $f$ in Theorem 3 with batch $M$, and let $f^\infty = \lim_{M\to\infty} f^M$. Then we have:*

- $f^M(\cdot) \leqslant q^1(\cdot), \forall M \in \mathbb{N}; f^\infty(\cdot) \leqslant q^1(\cdot).$
- $f^\infty(x_{1:n}) = h(x_n|x_{<n})[q^1(x_{1:n-1}) - f^\infty(x_{1:n-1})].$
- $\lim_{M\to\infty} \mathbb{E}[N_{rej}] = \sum_{n=1}^{T}(\mathbb{E}_{q^1}[\mathsf{TV}[q^1, p]] - \bar{\mathbb{E}}_{f^\infty}[\mathsf{TV}(q^1, p)]).$
- $M \to \infty, \mathbb{E}[N_{rej}] \nrightarrow 0.$ *This indicates increasing batch size to $\infty$ doesn't help.*

*Proof. First item:* Recall in the proof of Theorem 3, $f$ is defined as

$$f(x_{1:n}) :=\mathbb{P}^{\mathcal{A}}(\tilde{x}_n \text{ rej}, x_n|x_{1:n-1})q_n^1(x_{1:n-1})$$
$$=\mathbb{P}^{\mathcal{A}}(\tilde{x}_n \text{ rej}, x_n|x_{1:n-1})\mathbb{P}^{\mathcal{A}}(x_{1:n-1})$$
$$=\mathbb{P}^{\mathcal{A}}(\tilde{x}_n \text{ rej}, x_{1:n}) \leqslant \mathbb{P}^{\mathcal{A}}(x_{1:n}) = q^1(x_{1:n}),$$

where the second equality is due to Batch algorithm is unbiased (Theorem 3).

*Second item:* by

$$f(x_{1:n}) = h(x_n|x_{<n})q_n^1(x_{1:n-1}) - [h(x_n|x_{<n}) - (\prod_{m=1}^{M} \mathsf{TV}(q^m, p)(x_{<n}))q^{M+1}(x_n|x_{<n})]f(x_{1:n-1}),$$

since $\prod_{m=1}^{M} \mathsf{TV}(q^m, p)(x_{<n}) \to 0$ as $M \to 0$ (note $\mathsf{TV}(q^m, p)(x_{<n}) = 1$ iff there is no overlap between $q^m$ and $p$), it implies $f^\infty(x_{1:n}) = h(x_n|x_{<n})[q^1(x_{1:n-1}) - f^\infty(x_{1:n-1})]$.

*Third item:* Similar to the second item, it holds true via taking $M \to \infty$. For proving $\mathbb{E}[N_{\text{rej}}^\infty] > 0$, suppose $\mathbb{E}[N_{\text{rej}}^\infty] = 0$. Then it holds $q^1 \equiv f^\infty$. By the second item, this further implies $q^1 \equiv f^\infty = 0$, which is impossible.

*Fourth item:* We prove by contradiction. Suppose $\mathbb{E}[N_{\text{rej}}] \nrightarrow 0$, then by the first item and the third item this implies $q^1 \equiv f^\infty$. Plug this back to the second item, this further implies $f^\infty \equiv 0$, so $q^1 \equiv f^\infty \equiv 0$ is a contradiction ($q^1$ is a probability distribution)!

$\square$

# E  Proofs of Section 4

## E.1  Proof of Theorem 4

For the sampling scheme where the acceptance probability $b_n$ goes beyond $\min\{1, \frac{q_n(x|x_{1:n-1})}{p_n(x|x_{1:n-1})}\}$, there is a quality degradation for the output sequence as the algorithm is leaning towards the draft model (smaller model). In this case, the objective is to minimize the quality degradation via considering the TV distance

$$\min_{\mathcal{P}_n} \mathsf{TV}[\mathbb{P}_n^{\mathcal{A}}(\cdot|x_{1:n-1}), q_n(\cdot|x_{1:n-1})], \quad \forall x_{1:n-1} \in \mathcal{V}^{n-1}.$$

Under the above, via equation (14) the objective is equivalent to the following (note that according to Algorithm 2 $\mathbb{P}_n^{\mathcal{A}}(x_n|\text{draft token rejected}, x_{1:n-1}) = \mathcal{P}_n(x_n|x_{1:n-1})$ is an algorithmic choice)

$$
\begin{aligned}
\min_{\mathcal{P}_n} \quad & \frac{1}{2}\sum_x \left| q_n(x) - \min\{p_n(x), q_n(x) + \epsilon_n(x)\} - \mathcal{P}_n(x)\sum_{\tilde{x}}[p_n(\tilde{x}) - q_n(\tilde{x}) - \epsilon_n(\tilde{x})]_+ \right| \\
\text{s.t.} \quad & \sum_x \mathcal{P}_n(x) = 1, \quad \mathcal{P}_n(x) \geq 0, \ \forall x \in \mathcal{V}.
\end{aligned}
\tag{23}
$$

where we removed $x_{1:n-1}$ for notation simplicity, and recall again $\epsilon_n := b_n p_n - q_n$. When $\sum_{\tilde{x}}[p_n(\tilde{x}) - q_n(\tilde{x}) - \epsilon_n(\tilde{x})]_+ = 0$, the objective degenerates to the constant (in $\mathcal{P}_n$) $\mathsf{TV}(q_n, p_n)$. Therefore, for the rest of the section, we focus on the case where $\sum_{\tilde{x}}[p_n(\tilde{x}) - q_n(\tilde{x}) - \epsilon_n(\tilde{x})]_+ > 0$. We have the following Theorem that characterizes the solution of (23).

**Theorem 10** (Restatement of Theorem 4). *Suppose $\sum_x[p_n - q_n - \epsilon_n]_+(x) > 0$. Define*

$$A_n(x) := \frac{q_n(x) - \min\{p_n(x), q_n(x) + \epsilon_n(x)\}}{\sum_{\tilde{x}}[p_n(\tilde{x}) - q_n(\tilde{x}) - \epsilon_n(\tilde{x})]_+},$$

*and define the positive token set $A_+ = \{x \in \mathcal{V} : A_n(x) \geq 0\}$ and the negative token set $A_- = \{x \in \mathcal{V} : A_n(x) < 0\}$. Then the set of optimal distributions of objective (23) is characterized as*

$$\{\mathcal{P}_n^* : \mathcal{P}_n^*(x) = 0, \forall x \in A_-; 0 \leq \mathcal{P}_n^*(x) \leq A_n(x), \forall x \in A_+; \sum_x \mathcal{P}_n^*(x) = 1\},$$

*and the optimal value is*

$$Loss_{\mathsf{TV}}^*(b) = \frac{1}{2}\sum_x |q_n(x) - \min\{p_n(x), q_n(x) + \epsilon_n(x)\}| - \frac{1}{2}\sum_x[p_n(x) - q_n(x) - \epsilon_n(x)]_+.$$

**Remark 5.** *In the main context (Theorem 4) we define $A_n(x) := \frac{q_n(x) - b_n(x)p_n(x)}{\sum_{\tilde{x}}[1 - b_n(\tilde{x})]p_n(\tilde{x})}$ and $Loss_{\mathsf{TV}}^*(b) = \frac{1}{2}\sum_x |q_n(x) - b_n(x)p_n(x)| - \frac{1}{2}\sum_x(1 - b_n(x))p_n(x)$. This is equivalent to the above since $\epsilon_n := b_n p_n - q_n$.*

*Proof of Theorem 10.* In this case, note $\min\{p_n(x), q_n(x) + \epsilon_n(x)\} + [p_n(x) - q_n(x) - \epsilon_n(x)]_+ \equiv p_n(x)$, we have

$$\sum_x q_n(x) - \min\{p_n(x), q_n(x) + \epsilon_n(x)\} = \sum_{\tilde{x}}[p_n(\tilde{x}) - q_n(\tilde{x}) - \epsilon_n(\tilde{x})]_+. \tag{24}$$

By definition of $A_n$ then $\sum_x A_n(x) = 1$, and the original objective (23) can be equivalently written as

$$\min_{\mathcal{P}_n} \quad \frac{1}{2}\sum_x |A_n(x) - \mathcal{P}_n(x)|, \quad \text{s.t.} \sum_x \mathcal{P}_n(x) = 1, \quad \mathcal{P}_n(x) \geq 0, \ \forall x \in \mathcal{P}_n. \tag{25}$$

We now find the solution of objective (25) in two steps.

**Step1:** Let the positive token set $A_+ = \{x \in \mathcal{V} : A_n(x) \geq 0\}$ and the negative token set $A_- = \{x \in \mathcal{V} : A_n(x) < 0\}$, then any optimal $\mathcal{P}_n^*$ must satisfy $\mathcal{P}_n^*(x) = 0$ for all $x \in A_-$.

First, since $\sum_x A_n(x) = 1$, it implies $A_+ \neq \varnothing$. Suppose for some optimal $\mathcal{P}_n^*$, there exists $\bar{x} \in A_-$ such that $\mathcal{P}_n^*(\bar{x}) > 0$, then we show there exists $\check{x} \in A_+$ such that $A_n(\check{x}) > \mathcal{P}_n^*(\check{x})$. Suppose this is

not the case, i.e. $A_n(x) \leqslant \mathcal{P}_n^*(x) \; \forall x \in A_+$, then

$$1 = \sum_x A_n(x) = \sum_{x \in A_-} A_n(x) + \sum_{x \in A_+} A_n(x) \leqslant A_n(\bar{x}) + \sum_{x \in A_+} A_n(x)$$

$$< \sum_{x \in A_+} A_n(x) \leqslant \sum_{x \in A_+} \mathcal{P}_n^*(x) \leqslant \sum_{x \in \mathcal{V}} \mathcal{P}_n^*(x) = 1.$$

Contradiction! Hence, there exists $\check{x} \in A_+$ such that $A_n(\check{x}) > \mathcal{P}_n^*(\check{x})$.

Second, by $A_n(\check{x}) > \mathcal{P}_n^*(\check{x})$, $-\mathcal{P}_n^*(\bar{x}) < 0$, and triangular inequality, we have

$$| - \mathcal{P}_n^*(\bar{x})| + |A_n(\check{x}) - \mathcal{P}_n^*(\check{x})| > |A_n(\check{x}) - \mathcal{P}_n^*(\check{x}) - \mathcal{P}_n^*(\bar{x})|.$$

Note $A_n(\bar{x}) < 0$, the above is equivalent to

$$|A_n(\bar{x}) - \mathcal{P}_n^*(\bar{x})| + |A_n(\check{x}) - \mathcal{P}_n^*(\check{x})| > |A_n(\bar{x})| + |A_n(\check{x}) - \mathcal{P}_n^*(\check{x}) - \mathcal{P}_n^*(\bar{x})|. \qquad (26)$$

Now set

$$\mathcal{P}_n'(x) = \begin{cases} \mathcal{P}_n^*(x), & x \notin \{\bar{x}, \check{x}\}, \\ 0, & x = \bar{x}, \\ \mathcal{P}_n^*(\check{x}) + \mathcal{P}_n^*(\bar{x}), & x = \check{x}, \end{cases}$$

then apply (26) we have

$$\sum_x |A_n(x) - \mathcal{P}_n^*(x)| = \sum_{x \notin \{\bar{x}, \check{x}\}} |A_n(x) - \mathcal{P}_n^*(x)| + |A_n(\bar{x}) - \mathcal{P}_n^*(\bar{x})| + |A_n(\check{x}) - \mathcal{P}_n^*(\check{x})|$$

$$= \sum_{x \notin \{\bar{x}, \check{x}\}} |A_n(x) - \mathcal{P}_n'(x)| + |A_n(\bar{x}) - \mathcal{P}_n^*(\bar{x})| + |A_n(\check{x}) - \mathcal{P}_n^*(\check{x})|$$

$$> \sum_{x \notin \{\bar{x}, \check{x}\}} |A_n(x) - \mathcal{P}_n'(x)| + |A_n(\bar{x})| + |A_n(\check{x}) - \mathcal{P}_n^*(\check{x}) - \mathcal{P}_n^*(\bar{x})|$$

$$= \sum_{x \notin \{\bar{x}, \check{x}\}} |A_n(x) - \mathcal{P}_n'(x)| + |A_n(\bar{x}) - \mathcal{P}_n'(\bar{x})| + |A_n(\check{x}) - \mathcal{P}_n'(\check{x})| = \sum_x |A_n(x) - \mathcal{P}_n'(x)|.$$

This contradicts $\mathcal{P}_n^*$ is the optimal solution! Therefore, for any optimal $\mathcal{P}_n^*$, it holds $\mathcal{P}_n^*(x) = 0$ $\forall x \in A_-$.

**Step2:** We characterize the optimal solutions of the objective. Indeed, by Step1, any optimal solution $\mathcal{P}_n^*$ satisfies

$$\sum_x |A_n(x) - \mathcal{P}_n^*(x)| = \sum_{x \in A_-} |A_n(x) - \mathcal{P}_n^*(x)| + \sum_{x \in A_+} |A_n(x) - \mathcal{P}_n^*(x)|$$

$$= \sum_{x \in A_-} |A_n(x)| + \sum_{x \in A_+} |A_n(x) - \mathcal{P}_n^*(x)| \geqslant \sum_{x \in A_-} |A_n(x)| + |\sum_{x \in A_+} A_n(x) - \sum_{x \in A_+} \mathcal{P}_n^*(x)|$$

$$= \sum_{x \in A_-} |A_n(x)| + |\sum_{x \in A_+} A_n(x) - 1| = \sum_{x \in A_-} |A_n(x)| + \sum_{x \in A_+} A_n(x) - 1 = \|A_n\|_1 - 1.$$

The inequality becomes equality if and only if $A_n(x) - \mathcal{P}_n^*(x) \geqslant 0$. Finally, recall the definition of $A_n$ we receive the optimal value for the original objective is

$$\frac{1}{2} \sum_x |q_n(x) - \min\{p_n(x), q_n(x) + \epsilon_n(x)\}| - \frac{1}{2} \sum_x [p_n(x) - q_n(x) - \epsilon_n(x)]_+.$$

Replace $\epsilon_n$ by $b_n$ gives the desired result.

$\square$

**Remark 6.** *The general optimization should follow*

$$Loss_{\mathsf{TV}}^*(b_{1:T}) := \min_{\mathcal{P}} \mathsf{TV}[\mathbb{P}^{\mathcal{A}}(x_{1:T}), q(x_{1:T})], \quad \text{where } \mathcal{A} := (b_{1:T}, \mathcal{P}_{1:T}). \qquad (27)$$

*Meanwhile, our current analysis only considers the single token setting. We mention that solving the (27) is challenging as it corresponds to a high-dimensional discrete optimization with dimension $T$ and it might not have closed-form solutions in the general cases.*

### E.2 Proof of Theorem 5

*Proof.* For an algorithm $\mathcal{A}$ with the rejection probability $b(\cdot)$. Let $\tilde{x} \sim p(\cdot)$, then the rejection probability is computed as

$$\mathbb{P}(\text{reject}) = \sum_{\tilde{x}} \mathbb{P}(\text{reject}, \tilde{x}) = \sum_{\tilde{x}} \mathbb{P}(\text{reject}|\tilde{x})\mathbb{P}(\tilde{x}) = \sum_{\tilde{x}} (1 - b(\tilde{x}))p(\tilde{x}).$$

Also, from Theorem 4

$$\text{Loss}^*_{\text{TV}}(b) = \sum_x |q(x) - b(x)p(x)| - \sum_x (1 - b(x))p(x).$$

Next, we show

$$\sum_x |q(x) - b(x)p(x)| + \sum_x (1 - b(x))p(x) = \sum_x |p(x) - q(x)|.$$

Indeed, since $\min\{1, \frac{q(x)}{p(x)}\} \leqslant b(x) \leqslant 1, \ \forall x \in \mathcal{V}$, then $b(x)p(x) \geqslant \min\{p(x), q(x)\}$. Then we prove the following stronger claim

$$|q(x) - b(x)p(x)| + (1 - b(x))p(x) = |p(x) - q(x)|.$$

- If $q(x) \geqslant p(x)$, then $1 = \min\{1, \frac{q(x)}{p(x)}\} \leqslant b(x) \leqslant 1$ implies $b(x) = 1$, so the above is equivalent to $|q(x) - p(x)| = |p(x) - q(x)|$ is always true;

- If $q(x) < p(x)$, then $b(x)p(x) \geqslant \min\{p(x), q(x)\} = q(x)$. In this case

$$|q(x) - b(x)p(x)| + (1 - b(x))p(x) = b(x)p(x) - q(x) + (1 - b(x))p(x) = p(x) - q(x) = |p(x) - q(x)|.$$

This concludes the proof.

$\square$

# F   Numerical Simulation Details

To validate the correctness of our theory, we provide the numeric simulations that are displayed in Figure 4,2. We model the distribution $p_{1:T}$ and $q_{1:T}$ to be two non-stationary Markov Chains with 7 states/tokens. Instead of being $p(x_n|x_{1:n-1})$, for Markov Chain, the one step transition is Markovian with $p(x_n|x_{1:n-1}) = p(x_n|x_{n-1})$. We set the random seed to be 10. The prompt distribution $p_0$ is set to be Uniform distribution. For Figure 2, the true value is computed via Theorem 1,3 respectively, and solid line is computed by

$$N_{rej} := \frac{1}{N} \sum_{i=1}^{N} N_{rej}^i$$

and the shaded regions are error bars.

Below presents the simulation for horizon $T = 100$. Left panel of Figure 5 is the standard speculative decoding, the middle panel is batch speculative decoding with batch size $M = 2$, and Right panel shows the expected rejections with varying batch sizes $M$ computed from Theorem 1.

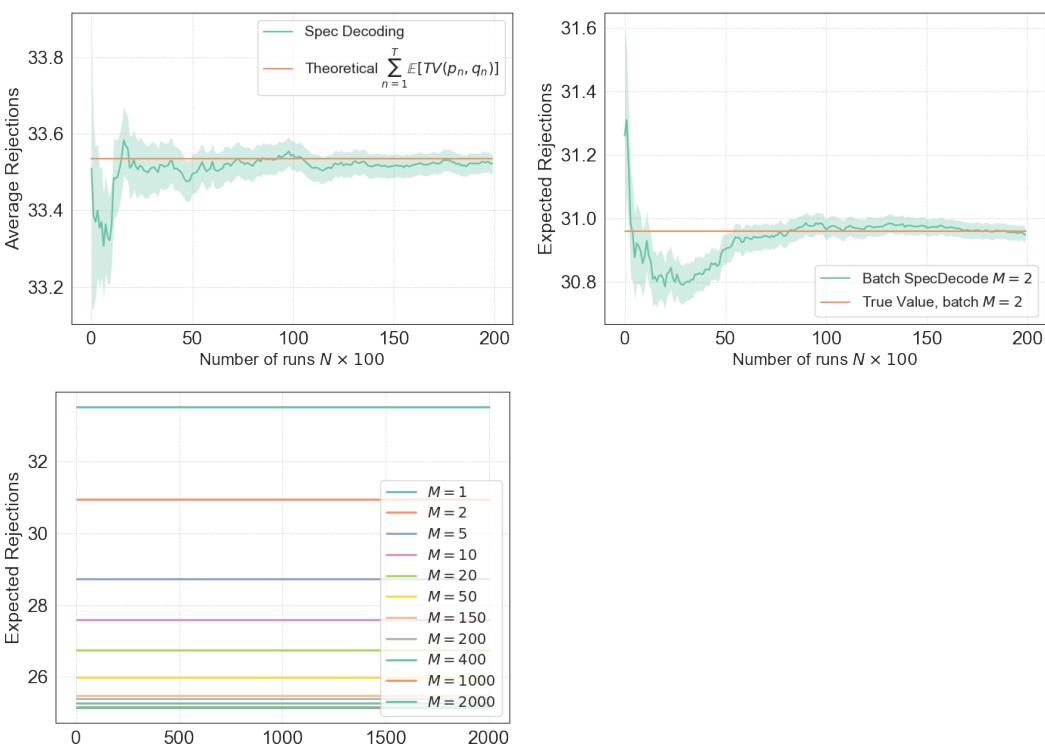

Figure 5: A simulation of (Batch) Speculative Decoding with horizon $T = 100$.

# G   Details for Experiment Section 4.2

Consider objective (1)

$$\text{Loss}^*_{\text{TV}}(b) := \min_{\mathcal{P}} \text{TV}[\mathbb{P}^{\mathcal{A}}, q], \quad where \ \mathcal{A} := (b, \mathcal{P}). \tag{28}$$

For any biased algorithm with the given acceptance probability $b > \min\{1, q/p\}$, we can rewrite:

$$b = \min\{1, \frac{q + \epsilon}{p}\}, \quad \text{for some } \epsilon > 0.$$

We consider the following two decoding options for $\mathcal{P}$:

- Baseline: select $\mathcal{P}$ to be suboptimal to (28), *i.e.* simply set $\mathcal{P} := q$, which is the target distribution;[10] We call this method `Decoding-UNO`.

- Set $\mathcal{P} := \mathcal{P}^*$ to be the optimal solution of (28), whose solution is presented in Theorem 4. We call this method `Decoding-OPT`.

**Measuring performance.** Instead of TV distance, we measure the quality via WinRate in our experiment. Concretely, for each prompt, we let `Decoding-UNO` and `Decoding-OPT` to generate responses independently, and use score models to compare whose generation has higher quality. We specify draft model $p$ as `pythia-70m` and target model $q$ as `pythia-2.8b` from EleutherAI [7]. We apply the score model to be `RM-Mistral-7B` or `GPT-4`. We test 200 prompts from `Alpaca-Farm-Eval` Dataset [13] with 500 responses/comparisons per prompt. For a given prompt, `Decoding-OPT` wins if for more than 250 comparisons, score model prefer its response[11] over `Decoding-UNO`. Then the WinRate for each method is computed as $\#wins/200$.

**Sanity Check for the experiment.** To validate having smaller distance w.r.t. the large model $q$ (`pythia-2.8b`) indicates higher performance, we preform the WinRate test for decoding via `pythia-70m` only against decoding via `pythia-2.8b` only. Table 2 shows that there is a significant performance gap between large model and small model, therefore validate the legitimacy of the experiment in Table 1.

| Method | RM-Mistral-7B | GPT-4 |
|---|---|---|
| `pythia-2.8b` | 64.5% | 69% |
| `pythia-70m` | 35.5% | 31% |

Table 2: WinRate for `pythia-2.8b` vs `pythia-2.8b`.

**Implementation detail for Table 1.** Concretely, we leverage `EleutherAI/pythia-2.8b` and `EleutherAI/pythia-70m` from HuggingFace. To perform speculative decoding, we specify `assistant_model=EleutherAI/pythia-70m` in the generation function for target model `EleutherAI/pythia-2.8b`.

Notice that for the biased speculative decoding with $b(x) = \min\{1, \frac{q(x)+\epsilon}{p(x)}\}$, $A(x)$ in Theorem 4 can equivalently expressed as $A(x) := \frac{\max\{q(x)-p(x), -\epsilon\}}{\sum_{\tilde{x}} \max\{q(\tilde{x})-p(\tilde{x}), -\epsilon\}}$, and we can select $\mathcal{P}^* := [A]_+$ (recall $[\cdot]_+$ in Section 2.1), which satisfies $\mathcal{P}^*|_{A_-}(\cdot) = 0; 0 \leqslant \mathcal{P}^*|_{A_+}(\cdot) \leqslant A(\cdot)$.

To implement `Decoding-UNO`, we modify the `_speculative_sampling` function in HuggingFace `transformers/generation/utils.py` file as follows[12] (where variable `eps_` is $\epsilon$ in Table 1). This is conducted in a single A100 GPU.

---

[10]To be rigorous, we mention we didn't choose $\mathcal{P} := [q - p]_+$ as the baseline since $\mathcal{P} := [q - p]_+$ might fall in the optimal solution sets of $\mathcal{P}^*$ defined in Theorem 4.

[11]We mention for `RM-Mistral-7B` it output values, then the response with higer value wins. For `GPT-4`, it outputs preference.

[12]Note the HuggingFace code uses $p$ as target model and $q$ as the draft model, which is different from us.

```
def _speculative_sampling(
    candidate_input_ids,
    ......,
):

    ......

    #####-----
    ## The following modification happens at Line 4727
    ## of the original file
    mode_ = 1 // mode_=1 denotes Decoding-OPT, else denotes Decoding-UNO
    eps_ = 0.1 // This is the epsilon in Table 1.

    _eps_ = eps_ * torch.ones(p_i.shape,dtype=torch.float32,\
    device=torch.device('cuda:0'))
    probability_ratio = (p_i + _eps_) / q_i

    #####-----

    ......

    if last_assistant_token_is_eos and n_matches == candidate_length:
        n_matches -= 1
        valid_tokens = new_candidate_input_ids[:, : n_matches + 1]
    else:
        n_matches = min(n_matches, max_matches)
        gamma = min(candidate_logits.shape[1], max_matches)
        p_n_plus_1 = p[:, n_matches, :]
        if n_matches < gamma:
            q_n_plus_1 = q[:, n_matches, :]
            ## The following modification happens at Line 4760
            ## of the original file
            if mode_ == 1:
                ## The following two lines compute A(x)
                p_prime = torch.clamp((p_n_plus_1 - q_n_plus_1), min= -eps_)
                p_prime.div_(p_prime.sum())
                ## The following two lines compute P* = [A]_+
                p_prime = torch.clamp(p_prime, min= 0)
                p_prime.div_(p_prime.sum())

            else:
                ## Baseline Decoding-UNO
                p_prime = q_n_plus_1
```

