# OpenReview forum: "A Theoretical Perspective for Speculative Decoding Algorithm"
_NeurIPS.cc/2024/Conference — NeurIPS 2024 poster_

### Official Review · Reviewer_qBZ2 · 2024-07-13

**Soundness:** 3
**Presentation:** 2
**Contribution:** 3
**Rating:** 6
**Confidence:** 2

**Summary:**

This paper presents a theoretical study on speculative decoding, an efficient inference method for large autoregressive models. It highlights practical implications, proposing a Pareto-optimal solution for the rejection-distribution bias tradeoff.

**Strengths:**

- The authors provide a robust theoretical foundation, illustrating the practical implications of speculative decoding, such as the improvement of rejection accuracy, which cannot be achieved by simply changing the acceptance probability.
 - The study explores the trade-offs between inference cost and quality degradation, supported by an optimization model. This analysis is valuable for practical applications.

**Weaknesses:**

- The main figure does not clearly communicate the core concept of speculative decoding. It might lead readers to believe that speculative decoding primarily addresses hallucination, which is not its main advantage.
 - The experimental results are not distinctly highlighted, and the authors do not explain how these results support their theoretical analysis. While the theoretical contributions are significant, the paper would benefit from more extensive empirical validation.

**Questions:**

NA

**Limitations:**

The results may not guarantee optimality in practical situations because real-world circumstances are more complex and varied than those considered in the theoretical analysis.

---

> ### Author Rebuttal · Authors · 2024-08-07
>
> We appreciate the reviewer for the positive judgement and the great questions. Here are the detailed responses.
>
> ***Q1:*** The main figure does not clearly communicate the core concept of speculative decoding. It might lead readers to believe that speculative decoding primarily addresses hallucination, which is not its main advantage.
>
> **Response:** Thank you very much, this is a very good suggestion! The purpose of Figure 1 is to show that, instead of direct decoding from the large model, speculative decoding will apply small model to decode and use large model as a verifier. We agree that without careful explanation the current figure might be confusing. We believe it is a easy fix and will modify it in the final revision.
>
> ***Q2:*** The experimental results are not distinctly highlighted, and the authors do not explain how these results support their theoretical analysis. While the theoretical contributions are significant, the paper would benefit from more extensive empirical validation.
>
> **Response:** We thank the reviewer for judging our theoretical contribution as significant. Regarding how the experimental results support our theory, our experiment 4.2 shows that, for different level of $\epsilon$  (over-accpetance), the optimal solution derived from Theorem 4 (```Decoding-OPT```) outperforms other suboptimal solutions (```Decoding-UNO```) consistently under the WinRate metric maresued by ```RM-Mistral-7B``` and GPT4. We will highlight this in the final version. Besides, regrading the paper would benefit from more extensive empirical validation, we conducted two extra experiments as explicitly asked by reviewer Ceup and reviewer GGsr. Here we summarize it for you.
>
> 1. In addition to experiment 4.2 where we validate the optimality of Theorem 4, we also compare the optimal algorithm (```Decoding-OPT```) with vanilla speculative decoding. Since the standard SD preserves the quality of the large model, the WinRate of our biased algorithm ```Decoding-OPT``` will decrease as the over-acceptance parameter $\epsilon$ goes large. On the other hand, the average runtime over 200 prompts for SD is much higher than ```Decoding-OPT``` in the third line of the table. This empirically validates ```Decoding-OPT``` provide a tradeoff between quality and efficiency against SD. For more detail please see our response to reviewer Ceup ***Q3.***
>
> | WinRate  | $\epsilon=0.1$ | $\epsilon=0.4$ | $\epsilon=0.8$ |
> | --- | --- | --- | --- |
> | Decoding-OPT | 48% | 42% | 35.5% |
> | SD | 52% | 58% | 64.5% |
> | Inference Acceleration rate: Time(SD)/ Time(Decoding-OPT)  | 1.54 | 2.97 | 6.32 |
> 2. Empirically, we also compare the Batch SD with vanilla SD and compute WinRate (measured by GPT4) over 200 prompts from Alpaca-Farm-Eval Dataset with 500 responses per prompt and the inference acceleration. The table shows that the quantity of batch algorithm is nearly the same as the vanilla SD, but the decoding is faster. However, when batch is very large (e.g. 10), the decoding speed can slow down (1.392<1.416) and this might due to batch processing cost, suggesting that using a large batch is not ideal in the real-world scenarios. For more detail please see our response to reviewer GGsr ***Q1.***
>
> | WinRate  | Batch = 2  | Batch = 4 | Batch = 8 | Batch = 10 |
> | --- | --- | --- | --- | --- |
> | Batch SD  | 48.5% | 49.5% | 49% | 49.5% |
> | Speculative Decoding | 51.5% | 50.5% | 51% | 50.5% |
> | Inference Acceleration rate: Time(SD)/ Time(Batch SD)  | 1.223 | 1.357 | 1.416 | 1.392 |

---

> ### Comment · Reviewer_qBZ2 · 2024-08-12
>
> Thank you for your response. I would like to remain the score.

---

> > ### Author Response · Authors · 2024-08-12
> > **Reply to Reviewer qBZ2**
> >
> > Dear reviewer qBZ2,
> >
> > Thank you for checking our rebuttal. Please let us know if you have any final questions.
> >
> > Best, Authors

---

### Official Review · Reviewer_GGsr · 2024-07-13

**Soundness:** 3
**Presentation:** 4
**Contribution:** 4
**Rating:** 6
**Confidence:** 5

**Summary:**

The paper presents a theoretical perspective on speculative sampling. Through Theorems 1 and 2, the authors demonstrate that the sampling method employed by speculative sampling is optimal and unbiased. Subsequently, Theorem 3 introduces a multi-candidate approach to enhance the acceptance rate of speculative sampling.

**Strengths:**

The writing is very clear, with takeaways provided under each theorem to explain the theory.

Theorems 1 and 2 are crucial for speculative sampling. In paper [23], the authors showed that speculative sampling is unbiased but did not prove its efficiency compared to other rejection sampling methods. The proof provided here is very important.

**Weaknesses:**

The experiments are not sufficient. I would like to see improvements in batch speculative sampling in real-world scenarios.

I am curious if batch speculative sampling can be combined with tree-style methods, e.g., [1] CaPE and [2] Medusa?

[1] Du, C., Jiang, J., Yuanchen, X., Wu, J., Yu, S., Li, Y., ... & You, Y. GliDe with a CaPE: A Low-Hassle Method to Accelerate Speculative Decoding. In Forty-first International Conference on Machine Learning.
[2] Cai, T., Li, Y., Geng, Z., Peng, H., Lee, J. D., Chen, D., & Dao, T. (2024). Medusa: Simple LLM inference acceleration framework with multiple decoding heads. In Forty-first International Conference on Machine Learning.

**Questions:**

see weakness

---

> ### Author Rebuttal · Authors · 2024-08-07
>
> We appreciate the reviewer for the positive comments and for the precise understanding of the theoretical contribution of our paper! Here are the detailed responses.
>
> ***Q1:*** I would like to see improvements in batch speculative sampling in real-world scenarios.
>
> Response: Thank you for the question. First, we want to mention that the main purpose of batch speculative sampling study is to understand its theoretical efficiency improvement. The key term we derived, the “Batch Improvement” in Theorem 3, is the net gain compared to the vanilla speculative decoding. When implementing the Batch algorithm 4 via the Markov Chain simulation, the plot in Figure 2 Right shows the improvement for the rejection rate as number of batches $M$ goes large. In addition, as the reviewer requested, we add the experiments for batch speculative decoding LLM decoding. Concretely, the speculative decoding of existing Hugging Face LLMs does not support batches, (i.e. the function ```_speculative_sampling``` in ```transformers/generation/utils.py```), so we modify the ```_speculative_sampling``` in page 30 by augmenting the related quantities such as ```candidate_input_ids``` with one extra dimension that contains batches. We then implement the batch algorithm 4 in the same function, and also modify the assisted_decoding function in transformers/generation/utils.py. We compare the Batch SD with vanilla SD and compute WinRate (measured by GPT4) over 200 prompts from Alpaca-Farm-Eval Dataset with 500 responses per prompt and the inference acceleration. The draft model is still ```pythia-70m``` and the target model is still ```pythia-2.8b```. The table shows that the quantity of batch algorithm is nearly the same as the vanilla SD, but the decoding is faster. However, when batch is very large (e.g. 10), the decoding speed can slow down (1.392<1.416) and this might due to batch processing cost, suggesting that using a large batch is not ideal in the real-world scenarios. Running the experiments in the table spent 70+ A100 GPU hours.
>
> | WinRate  | Batch = 2  | Batch = 4 | Batch = 8 | Batch = 10 |
> | --- | --- | --- | --- | --- |
> | Batch SD  | 48.5% | 49.5% | 49% | 49.5% |
> | Speculative Decoding | 51.5% | 50.5% | 51% | 50.5% |
> | Inference Acceleration rate: Time(SD)/ Time(Batch SD)  | 1.223 | 1.357 | 1.416 | 1.392 |
>
> ***Q2:*** I am curious if batch speculative sampling can be combined with tree-style methods, e.g., [1] CaPE and [2] Medusa?
>
> Response: Thank you for the in-depth question! This is an interesting aspect since the current Batch SD (Algorithm 4) is mostly suitable for theoretical study and might not be optimal for practical application. For instance, the batch rejection scheme in Algorithm 4 is to simply iterate over the next batch when the previous batches are rejected. It does not distinguish the unique characterizations of the batch candidate tokens. In this sense, CAPE [1], which uses uncertainty metric such as confidence score to adaptively choose the next token could further improve the chance of the token being accepted. Besides, Batch Algorithm 4 represents a simple parallel tree structure (Figure 3 Left), modify it with some tailor-designed tree-based methods such as tree-based attention in Medusa [2] is very likely to make batch decoding more efficient. We promise to add the discussions on [1],[2] in the final revision and leave how to make it work as future work.

---

> > ### Comment · Reviewer_GGsr · 2024-08-12
> >
> > Due to this method can not be used in the real setting like tree spec, I still maintain my initial score.

---

> ### Author Response · Authors · 2024-08-12
> **Reply to Reviewer GGsr**
>
> Dear reviewer GGsr,
>
> Thank you for checking our rebuttal. To be academically rigorous, we do agree with the reviewer that our proposed batch algorithm 4 is different from the existing tree spec such as SpecInfer [Miao et al.], CAPE and Medusa since our setup (alg4) is suitable for the theoretical understanding (batch improvement in Theorem 3 and Figure 2 Right) for the efficiency of spec decoding, which is the main focus of the paper and this theory is lacking in the existing batch tree spec algorithms. We promise to discuss it carefully in the revision and will mention the limitation that it remains unclear on how to combine batch algorithm 4 with other tree-spec algorithms as you suggested. Thank you again for the great point.
>
> Best, Authors

---

### Official Review · Reviewer_XUMm · 2024-07-13

**Soundness:** 3
**Presentation:** 3
**Contribution:** 3
**Rating:** 6
**Confidence:** 2

**Summary:**

The author aim to develop theoretical understanding of speculative decoding. The authors assume that given a large and small model participating in speculative decoding, the computation complexity of the small model is negligible. Under this assumption, they characterize the expected rejection rate of speculative decoding. They show that this bound depends on the total variation distance between the generations from small and large model. Next, the authors show that spectral decoding gives optimal rejection bounds in class of all rejection based methods. Motivated by recent works analyzing batch speculative decoding, where the rejection is done only if M tokens are rejected from a given sample. Finally, given an acceptance probability, the authors show an optimal solution solution to the total variation loss between the distributions of large model and the one found by speculative decoding. This objective changes linearly with the rejection probability. This provides insights on selecting the optimal value of rejection threshold as per requirement. The presented theoretical results are backed up with appropriate experiments validating them.

**Strengths:**

1) The theoretical analysis presented by the authors provide several interesting insights about the inference efficiency observed by spectral decoding.

2) All the results are backed up with simulation experiments, which strengthen the results presented in the paper.

**Weaknesses:**

1) It is not completely clear, why making the assumption about negligible compute of the small model is not a strong assumption. Since the small model needs to generate the tokens autoregressively therefore even though its single pass could be small as compared to the larger model but it the context length is high i.e. several autoregressive passes are made, the compute of small model might not be negligible. It would be great if the authors can provide some empirical evidence to justify this assumption.

2) It would have been great if the authors provided evidence using real world models in support of their theory. Although, this is not a major weakness, but authors should consider it in the camera ready version.

**Questions:**

I request the authors to kindly address the questions in the weaknesses section.

**Limitations:**

Yes, the authors have addressed the limitations.

---

> ### Author Rebuttal · Authors · 2024-08-07
>
> We appreciate the reviewer for the positive judgement and the detailed feedback. Here are the detailed responses.
>
> ***Q1:*** It is not completely clear, why making the assumption about negligible compute of the small model is not a strong assumption. Since the small model needs to generate the tokens autoregressively therefore even though its single pass could be small as compared to the larger model but it the context length is high i.e. several autoregressive passes are made, the compute of small model might not be negligible.
>
> **Response:** Thank you for the question about our assumption. First, we fully agree that, in practice, all draft models cost some time. However, the main study goal of this paper is to understand the theoretical/probabilistic property of speculative decoding, and the cost-negligible assumption is an abstraction for the theoretical cleanliness. The assumption we made highlights that the cost is only negligible compared to the large model, rather than assuming it has zero cost. These two have fundamental differences since cost-negligible means there is some cost, and the cost is theoretically abstracted in the notation O(1) (see second part of Assumption 1) when compared to some costly large model. This is some known concept summarized in [Leviathan et al., Jie Ou et al.]. Empirically, this is also very meaningful. For instance, when measuring the inference time over 200 responses of Alpaca-Farm-Eval for model ```pythia-70m``` and model ```pythia-12b``` separately, the ratio ```Time(pythia-12b)/Time(pythia-70m)```$\approx$ 12.8 for T=128 tokens, and for T=1028tokens, the ratio$\approx $18.3 is even larger (this is not surprising since the decoding time for auto-regressive model is super-linear in its decoding length and large model would be even slower compard to the small models). In this case, the decoding time for ```pythia-70m``` is not cost zero, but is smaller than ```pythia-12b``` by magnitude, therefore cost-negligible.
>
> ***Q2:*** It would have been great if the authors provided evidence using real world models in support of their theory. Although, this is not a major weakness, but authors should consider it in the camera ready version.
>
> **Response:** Thank you for the kind question. We believe the models we used in experiment 4.2 (```pythia-70m``` and ```pythia-12b```) are real-world models, and our experiment 4.2 shows that, for different level of $\epsilon$  (over-accpetance), the optimal solution derived from Theorem 4 (```Decoding-OPT```) outperforms other suboptimal solutions (```Decoding-UNO```) consistently under the WinRate metric maresued by ```RM-Mistral-7B``` and GPT4. We will highlight this in the final version. Besides, we conducted two extra experiments as explicitly asked by reviewer Ceup and reviewer GGsr. Here we summarize it for you.
>
> 1. In addition to experiment 4.2 where we validate the optimality of Theorem 4, we also compare the optimal algorithm (```Decoding-OPT```) with vanilla speculative decoding. Since the standard SD preserves the quality of the large model, the WinRate of our biased algorithm ```Decoding-OPT``` will decrease as the over-acceptance parameter $\epsilon$ goes large. On the other hand, the average runtime over 200 prompts for SD is much higher than Decoding-OPT in the third line of the table. This empirically validates Decoding-OPT provide a tradeoff between quality and efficiency against SD. For more detail please see our response to reviewer Ceup ***Q3.***
>
> | WinRate  | $\epsilon=0.1$ | $\epsilon=0.4$ | $\epsilon=0.8$ |
> | --- | --- | --- | --- |
> | Decoding-OPT | 48% | 42% | 35.5% |
> | SD | 52% | 58% | 64.5% |
> | Inference Acceleration rate: Time(SD)/ Time(Decoding-OPT)  | 1.54 | 2.97 | 6.32 |
> 2. Empirically, we also compare the Batch SD with vanilla SD and compute WinRate (measured by GPT4) over 200 prompts from Alpaca-Farm-Eval Dataset with 500 responses per prompt and the inference acceleration. The table shows that the quantity of batch algorithm is nearly the same as the vanilla SD, but the decoding is faster. However, when batch is very large (e.g. 10), the decoding speed can slow down (1.392<1.416) and this might due to batch processing cost, suggesting that using a large batch is not ideal in the real-world scenarios. For more detail please see our response to reviewer GGsr ***Q1.***
>
> | WinRate  | Batch = 2  | Batch = 4 | Batch = 8 | Batch = 10 |
> | --- | --- | --- | --- | --- |
> | Batch SD  | 48.5% | 49.5% | 49% | 49.5% |
> | Speculative Decoding | 51.5% | 50.5% | 51% | 50.5% |
> | Inference Acceleration rate: Time(SD)/ Time(Batch SD)  | 1.223 | 1.357 | 1.416 | 1.392 |

---

> > ### Comment · Reviewer_XUMm · 2024-08-13
> >
> > I thank the authors for their rebuttal. I acknowledge that I have read their reply. This increases my confidence in this work.

---

### Official Review · Reviewer_Ceup · 2024-07-13

**Soundness:** 2
**Presentation:** 3
**Contribution:** 2
**Rating:** 5
**Confidence:** 4

**Summary:**

This paper provides detailed analysis to speculative decoding and batch speculative decoding. The conclusions of the paper are: (1) speculative decoding is unbiased and it shows the expected rejection rate; (2) speculative decoding has the lowest rejection rate in all the unbiased algorithm that belongs to the familty defined in Algorithm 2; (3) batch speculative decoding has lower rejection rate than speculative decoding; (4) it analyzes the trade-off between efficiency and effectiveness of the family of algorithm defined in Algorithm 2.

**Strengths:**

1. The paper provides comprehensive theoretical analysis.

2. The findings in Theorem 4 and 5 are interesting.

3. The paper is easy to understand.

**Weaknesses:**

1. Although the paper provides lots of theoretical analysis. But I find only Theorem 4 and 5 are somewhat interesting. Theorem 1 is already derived in the original speculative decoding paper. For Theorem 2, although speculative decoding is proven to be optimal in the family of algorithms defined in Algorithm 2, but I don't think there are a lot of existing algorithms can be formulated in Algorithm 2. In fact, is there any algorithm that belongs to Algorithm 2 and is unbiad and it not speculative decoding? For Theorem 3, the finding that batch speculative decoding has lower rejection rate than vanilla speculative decoding is not surprising.

2. Although Theorem 4 and Theorem 5 are interesting, it only solves half of the problem: given b, what should P be. It would be better if the authors could also discuss the design of b.

3. I think the paper can also be improved if the authors could summarize a new speculative algorithm from Theorem 4 and 5 and running experiments to compare with vanilla speculative decoding.

**Questions:**

see weakness above

**Limitations:**

see weakness above

---

> ### Author Rebuttal · Authors · 2024-08-07
>
> We thank the reviewer for providing detailed feedback. We have read your comments carefully and below are our detailed responses.
>
> **Q1:** ---- Although ... Theorem 1 derived in the original speculative decoding paper. For Theorem 2, is there any algorithm that belongs to Algorithm 2 and is unbiased and it not speculative decoding? For Theorem 3, ... is not surprising. ----
>
> **Response:** Thanks, but we respectfully disagree that only Theorem 4 and 5 are interesting, and we believe Theorem 1-3 have their own and new merits for understanding the theoretical properties for speculative decoding, which we detail below.
>
> **On Thm 1:** The key distinction between our Thm1 and [Leviathan et al.] is that their derivation assumes i.i.d. generation of LLMs (see “If we make the simplifying assumption that the beta’s are i.i.d.” under their Definition 3.1), which almost never holds in practice since the distribution of the next token depends on the past generations which is not i.i.d. From the theoretical aspect, our metric $T/\sum_T E_{x_{1:n-1}~q}[TV(p_n,q_n)(\cdot|x_{1:n-1})]$ captures the sequential dependence via the conditional dependence on $x_{1:n-1}$. In this sense, our result is a precise theoretical measurement of sequence (that have dependence between tokens), and the result in [Leviathan et al.] only holds for a single token or a fully i.i.d sequence which is not practical (also see our Remark2).
>
> **On Thm 2:** The significance of our optimality result is that it tells practitioners no improvement can be made over SpecDecoding if no other information can be leveraged. This is a theoretical guidance that can save time for practitioners who want to improve the efficiency of SpecDecoding. In addition, there are many algorithms that belongs to Algorithm 2 and is unbiased (i.e. in $\mathcal{F}$) but is not speculative decoding. This is summarized in our **Corollary 1** in appendix. For any $b_n$ satisfies $b_n\leq \min\\{1,\frac{q_n}{p_n}\\}$ for all $n\in[T]$, we can find an Algorithm 2 that is unbiased. This exactly provide the intuition why is SpecDecoding is optimal: if an Algorithm 2 wants to be unbiased, it needs to have smaller acceptance probability $b_n$ than speculative decoding $\min\{1,\frac{q_n}{p_n}\}$. We were not able to include this in the main paper due to space constraint, and we will add a comment in the final revision.
>
> **On Thm3:** We agree with the reviewer’s intuition that batch algorithm can reduce the rejection rate, however, such a reduction the rejection rate is only observed empirically and a precise theoretical understanding of how much it can improve given the structure $p_n,q_n $ remains unknown. For instance, the nice work SpecInfer [Miao et al.] only prove the unbiasedness but there is no guarantees for decoding efficiency. To this end, we design a different parallel tree structure (Line 213) for batch setting. In order to derive the Batch Improvement term in Theorem 3, we design the novel intermediate quantity $f(x_{1:n})$ in D.1 with recursive computation for solving it, and the numerical simulation validates our theory. The significance of our batch theory is that: their is a scaling law for even using huge number of batches, meaning that the rejection rate won’t goes to zeros even the number of batch goes to infinity (see Figure 2 Right and D.2 in appendix). We believe all those insights are new.
>
> **Q2:** ---- Although Theorem 4 and Theorem 5 are interesting, it only solves half of the problem: given b, what should P be. It would be better if the authors could also discuss the design of b. ----
>
> **Response:** Thanks. This is a good question. The design of $b$ is more heuristic since with higher $b$ (accpeatance probability), there will be less rejections and the decoding is faster, but it will suffer quality loss according Theorem 4. It is a matter of how much response quality one is willing to suffer for gaining better efficiency. In our simulations and experiments in 4.2, we specify $b(x)=\min \\{1, \frac{q(x)+\epsilon}{p(x)}\\} $ for $\epsilon$ ranging from 0 to 1. We will add more discussions in the final revision.
>
> **Q3:** ----if the authors could summarize a new speculative algorithm from Theorem 4 and 5 and running experiments to compare with vanilla speculative decoding.----
>
>
> **Response:** Thanks. Indeed, the new algorithm that summarizes Theorem 4&5 is exactly Algorithm 2. Concretely, with the choice of $b(x)=\min \\{1, \frac{q(x)+\epsilon}{p(x)}\\}$, we specify the $\mathcal{P}_t $ in Algorithm 2 to be the optimal one $\mathcal{P}_t^\star$ which is defined in Line 756 in appendix and the code implementation is in page 30 starting from ```if mode_ ==1```). We compare our optimal algorithm ```Decoding-OPT``` with a suboptimal algorithm ```Decoding-UNO``` to show higher WinRate across different $\epsilon $s in section 4.2. In addition to that, as the reviewer requested, we conduct extra experiment below to compare the performance of ```Decoding-OPT``` and vanilla SD. The draft model is still ```pythia-70m``` and the target model is still ```pythia-2.8b```. The following table is the WinRate measured by GPT-4. We test 200 prompts from ```Alpaca-Farm-Eval``` Dataset with 500 responses per prompt. Since the standard SD preserves the quality of the large model, the WinRate of our biased algorithm ```Decoding-OPT``` will decrease as the over-acceptance parameter $\epsilon$ goes large. On the other hand, the average runtime over 200 prompts for SD is much higher than ```Decoding-OPT``` in the third line of the table. This empirically validates ```Decoding-OPT``` provides a tradeoff between quality and efficiency against SD. Running the experiments in the table spent 50+ A100 GPU hours.
>
> | WinRate  | $\epsilon=0.1$ | $\epsilon=0.4$ | $\epsilon=0.8$ |
> | --- | --- | --- | --- |
> | Decoding-OPT | 48% | 42% | 35.5% |
> | SD | 52% | 58% | 64.5% |
> | **Inference Acceleration rate:** Time(SD)/ Time(Decoding-OPT)  | 1.54 | 2.97 | 6.32 |

---

> > ### Comment · Reviewer_Ceup · 2024-08-11
> >
> > I appreciate the author’s efforts in providing the rebuttal. After reading the rebuttal and other reviewer’s opinions, I decided to raise my score.

---

> > > ### Author Response · Authors · 2024-08-12
> > > **Reply to reviewer Ceup**
> > >
> > > Dear reviewer Ceup,
> > >
> > > Thank you for spending time reading our rebuttal. We authors are still available here to answer your questions in the next two days in case you have any last-minute questions.
> > >
> > > Best, Authors

---

### Decision · Program_Chairs · 2024-09-25

**Decision:**

Accept (poster)

**Comment:**

This paper provides detailed analysis of speculative decoding and batch speculative decoding. The conclusions of the paper are: (1) speculative decoding is unbiased and it shows the expected rejection rate; (2) speculative decoding has the lowest rejection rate in all the unbiased algorithm that belongs to the family defined in Algorithm 2; (3) batch speculative decoding has lower rejection rate than speculative decoding; (4) it analyzes the trade-off between efficiency and effectiveness of the family of algorithm defined in Algorithm 2.
The paper is well written and easy to understand. The theoretical analysis presented by the authors provide several interesting insights about the inference efficiency observed by speculative decoding. All the results are backed up with simulation experiments, which strengthen the results presented in the paper. After the author rebuttal, the only major remaining ask would be to see improvements in batch speculative sampling in real-world scenarios which is currently impossible.